# High-resolution (0.05°×0.05°) NO$_x$ emissions in the Yangtze River Delta inferred from OMI

Hao Kong[1], Jintai Lin[1], Ruixiong Zhang[1*], Mengyao Liu[1], Hongjian Weng[1], Ruijing Ni[1], Lulu Chen[1], Jingxu Wang[1], Yingying Yan[2], Qiang Zhang[3]

[1]Laboratory for Climate and Ocean-Atmosphere Studies, Department of Atmospheric and Oceanic Sciences, School of Physics, Peking University, Beijing 100871, China

[2]Department of Atmospheric Science, School of Environmental Sciences, China University of Geosciences, Wuhan 430074, China

[3]Ministry of Education Key Laboratory for Earth System Modeling, Department of Earth System Science, Tsinghua University, Beijing 100084, China

*Correspondence to*: Jintai Lin (linjt@pku.edu.cn)

[*] Now at: School of Earth and Atmospheric Sciences, Georgia Institute of Technology, Atlanta, GA, USA

## Abstract

Emission datasets of nitrogen oxides (NO$_x$) at high horizontal resolutions (e.g., 0.05°×0.05°) are crucial for understanding human influences at fine scales, air quality studies, and pollution control. Yet high-resolution emission data are often missing or contain large uncertainties especially for the developing regions. Taking advantage of long-term satellite measurements of nitrogen dioxide (NO$_2$), here we develop a computationally efficient method of estimating NO$_x$ emissions in major urban areas at the 0.05°×0.05° resolution. The top-down inversion method accounts for the nonlinear effects of horizontal transport, chemical loss, and deposition. We construct a model called PHLET (2-dimensional Peking University High-resolution Lifetime-Emission-Transport model), its adjoint model (PHLET-A), and a

Satellite Conversion Matrix approach to relate emissions, lifetimes, simulated $NO_2$, and satellite $NO_2$ data. The inversion method is applied to summer months of 2012–2015 in the Yangtze River Delta area (YRD, 118°E-123°E, 29°N-34°N), a major polluted region of China, using the POMINO $NO_2$ vertical column density product retrieved from the Ozone Monitoring Instrument. A systematic analysis of inversion errors is performed, including using an Observing System Simulation Experiment-like test. Across the YRD area, the summer average emissions obtained in this work range from 0 to 15.3 kg $km^{-2}$ $h^{-1}$, and the lifetimes (due to chemical loss and deposition) from 0.6 to 3.3 h. Our emission dataset reveals fine-scale spatial information related to nighttime light, population density, road network, maritime shipping, and land use (from a Google Earth photo). We further compare our emissions with multiple inventories. Many of the fine-scale emission structures are not well represented or not included in the widely used Multi-scale Emissions Inventory of China (MEIC).

## 1. Introduction

Nitrogen oxides ($NO_x$ = NO + $NO_2$) are a main precursor of particulate matter, ozone, and other atmospheric pollutants. $NO_x$ strongly influence the atmospheric oxidative capacity, affect the climate, and are toxic to many organisms. $NO_x$ are emitted from natural and anthropogenic sources (Lin, 2012). Over the past decade, China has experienced rapid growth in the Gross Domestic Product (GDP, by 8.3% $a^{-1}$ on average from 2008 to 2017), fossil fuel consumption (by 5.5% $a^{-1}$ from 2007 to 2015), and urbanization (National Bureau of Statistics of China, http://data.stats.gov.cn/). These socioeconomic changes have been accompanied by a rapid change in $NO_x$ emissions in the urban and surrounding areas. With the large and continuously increasing urban population and motor vehicles, $NO_x$ pollution is particularly severe in large cities such as Beijing and Shanghai (Barnes and Rudziński, 2013; Lin et al., 2016). Many coastal cities like Shanghai have also experienced enormous growth in the shipping business. Therefore, pollution along the coastal line has become a serious problem associated with the growth of global economic trade; and emissions from seaborne transport play an increasingly important role in the global air pollution (Fu et al., 2017). Understanding the urban pollution and its environmental impacts requires accurate

quantitative knowledge of $NO_x$ emissions at a very high horizontal resolution (e.g., 0.05°×0.05°), which is typically lacking especially for the developing countries.

Gridded bottom-up emission inventories typically use spatial proxies (like population and GDP) to allocate provincial-level emission values, which are derived from activity statistics and emission factor data, to individual locations (Zhao et al., 2011; Janssens-Maenhout et al., 2015; Zhao et al., 2015). Such a gridding method may lead to large uncertainties at high resolutions (Geng et al., 2017), because the mismatch between proxies and emissions becomes more significant and emitting facilities are harder to allocate accurately as the resolution increases (Zheng et al., 2017). For a small area, emission factors and activity data of the major sources can be collected by on-site surveys to allow construction of a high-resolution inventory (Zhao et al., 2015; Granier et al., 2019), such as Zhao et al. (2015) for Nanjing. However, on-site surveys are extremely time consuming and resource demanding, and therefore difficult to be applied to a large domain in a timely manner.

Top-down inversion using satellite retrieval products of tropospheric vertical column densities (VCDs) of nitrogen dioxide ($NO_2$) is a widely used independent estimate of $NO_x$ emissions (Martin et al., 2003; Stavrakou et al., 2008; Lin et al., 2010; Mijling and van der A, 2012; Gu et al., 2014; Beirle et al., 2015; Miyazaki et al., 2016; Ding et al., 2017b). Top-down inversion typically provides the total emission data, although emissions from individual sources can be further derived by integrating a priori data (often from bottom-up inventories) about source-specific information such as diurnal and seasonal variabilities (e.g., Lin et al., 2010; Lin, 2012) and spatial variabilities (Timmermans et al., 2016).

The traditional top-down methods based on local mass balance (LMB) or its variants assume a weak effect of horizontal transport (Martin et al., 2003;Lamsal et al., 2011;Lin, 2012;Gu et al., 2014;Boersma et al., 2015). These algorithms work relatively well at low resolutions (> 50 km) given the relatively short lifetime of $NO_x$ (hours to 1 day), but may introduce large uncertainties when applied to higher resolutions – for example, emissions in the rural-urban fringe zone cannot be identified accurately. The Adjoint Model and Kaman Filter methods better account for horizontal transport, although their applicability is

limited by expensive computational costs. These more sophisticated methods have often been applied to relatively short time periods (e.g., Gu et al., 2016 for one month), small spatial domains (e.g., Tang et al., 2013 in Texas), and/or at coarse horizontal resolutions (e.g., Miyazaki et al., 2012 at T41 grid, i.e.; ~2.8°, and Stavrakou et al., 2008 at 5°×5°). Top-down estimates can be further combined with bottom-up inventories and spatial proxies to increase the spatial resolution, such as from 0.25°×0.25° in the DECSO derived emissions to 0.01°×0.01° for 2014 during the MarcoPolo Project (Hooyberghs et al., 2016; Timmermans et al., 2016) and similar inventories over Qatar and South Africa (Maiheu and Veldeman, 2013). The LMB, Adjoint Model and Kaman Filter approaches normally use 3-dimentional chemical transport models (CTMs) to relate emissions to VCDs. CTM-based studies typically provide an estimate of the overall model error, although Lin et al. (2012) and Stavrakou et al. (2013) present errors in the individual model processes (e.g., key chemical reactions and meteorological parameters). A computationally low-cost method for space-based high-resolution (0.05°×0.05°) $NO_x$ emission estimate will be helpful for understanding the urban pollution and its trends and variability.

This study presents a computationally low-cost space-based top-down approach to construct high-resolution $NO_x$ emission inventories for urban and surrounding areas. The approach is applied to the Yangtze River Delta (YRD) area (118°E-123°E, 29°N-34°N, which includes Shanghai, Nanjing, Hangzhou and 15 other cities) on a 0.05°×0.05° grid, using the POMINO $NO_2$ VCD data retrieved from the Ozone Monitoring Instrument (OMI). We derive the average $NO_x$ emissions for the summer months (June, July, and August) of 2012–2015. We construct a model called PHLET (2-dimensional Peking University High-resolution Lifetime-Emission-Transport) and its adjoint model (PHLET-A) to facilitate the emission estimate. The lifetimes of $NO_x$ are estimated as well, in order to account for the nonlinear $NO_x$ chemistry.

Section 2 presents the data and method for top-down inversion of high-resolution $NO_x$ emissions. Inversion uncertainties are analyzed explicitly. Section 3 presents spatial distributions of $NO_2$ VCDs, the derived local net sources (which are used subsequently to derive $NO_x$ emissions and lifetimes), $NO_x$ lifetimes, and their uncertainties. Section 4 analyzes the top-down emission data estimated here, including

comparisons with spatial proxies (population density, night light brightness, power plant locations, road network, marine shipping routes, and a Google Earth photo for land use indication), the Multi-scale Emissions Inventory of China (MEIC) (Zheng et al., 2014; Liu et al., 2015), the DECSO top-down emissions (Mijling et al., 2013; Ding et al., 2017a), and the MarcoPolo emissions (Hooyberghs et al., 2016; Timmermans et al., 2016). Section 5 tests our inversion method by applying it to the $NO_2$ VCDs simulated by the GEOS-Chem CTM. Section 6 concludes the study.

## 2. Data and Method

### 2.1. A general framework to retrieve $NO_x$ emissions at a high resolution

The high-resolution $NO_x$ emission retrieval framework consists of multiple steps, as illustrated in the flowchart (Fig. 1). First (Sect. 2.2), the POMINO $NO_2$ VCD data over summer 2012–2015 are averaged on a $0.05°\times0.05°$ grid, using a special oversampling technique that preserves the finest spatial information possible. A Satellite Conversion Matrix (SCM), which will be applied to PHLET simulated $NO_2$ VCDs at the second step, is also calculated based on the OMI pixel parameters (i.e. corner co-ordinates).

Second (Sect. 2.3), the PHLET model is constructed to simulate the local net source (i.e., emission – loss) and horizontal transport of $NO_2$ VCDs on the $0.05°\times0.05°$ grid. The SCM is then applied to PHLET simulated VCDs to mimic how each satellite pixel averages the spatial distribution of $NO_2$, in order to ensure the spatial sampling consistency between PHLET and POMINO. This process is needed because satellite pixels represent the $NO_2$ spatial distribution at a coarser (than PHLET) resolution with irregular shapes of individual pixels.

Third (Sect. 2.4), the PHLET-A adjoint model is constructed to, together with PHLET and POMINO VCDs, derive the local net source at each $0.05°\times0.05°$ grid cell. We construct a cost function to quantify the difference between the distribution of POMINO VCDs and that simulated by PHLET at the $0.05°\times0.05°$ grid. The inversion process to derive the local net sources is equivalent to minimization of the cost function.

Finally (Sect. 2.5), the emission and lifetime of $NO_x$ at each $0.05° \times 0.05°$ grid cell is derived from the local net source term, by fitting a formula for the nonlinear relationship between lifetimes and VCDs. The formula is assumed to be fixed, i.e., the relationship is applicable to all grid cells within the small study domain.

Furthermore, a rigorous error analysis for the framework and models is conducted (Sect. 2.6). This analysis is complemented by an test based on the GEOS-Chem simulated distribution of $NO_2$ VCDs (Sect. 5).

Our inversion method explicitly accounts for horizontal transport and the nonlinear relationship between lifetimes and $NO_2$ VCDs. With a few reasonable assumptions, the method is computationally efficient,
suitable for speedily conducting high-resolution emission estimates in multiple areas and across a long time period (2012-2015 in this study). Both PHLET and PHLET-A are numerically solved based on FEniCS, a popular open source solver (Farrell et al., 2012; Funke and Farrell, 2013; Alnaes et al., 2015). With one computational core (Intel® Xeon® Gold 6130 CPU @ 2.10GHz), derivation of $NO_x$ emissions over the YRD here takes less than one hour after necessary input data are prepared. Applying the
framework to multiple areas would take a similar amount of time by using one computational core for each area.

## 2.2. Tropospheric $NO_2$ VCDs retrieved from OMI

OMI is a UV/VIS nadir solar backscatter spectrometer on board the Aura satellite (Levelt et al., 2006). OMI provides daily global coverage. Each complete swath of OMI consists of 60 ground pixels, the sizes
of which increase from $13 \text{ km} \times 24 \text{ km}$ at nadir to about $40 \text{ km} \times 150 \text{ km}$ at the swath edge in accordance to the view zenith angle (VZA) from $0°$ to $57°$ (de Graaf et al., 2016).

We use Level-2 tropospheric $NO_2$ VCD data from POMINO (Peking University Ozone Monitoring Instrument $NO_2$ product) (Lin et al., 2014; Lin et al., 2015a). As described in detail in Lin et al. (2014; 2015), POMINO is an OMI-based regional $NO_2$ product that includes a number of important features.

Briefly, POMINO adopts the tropospheric slant column density (SCD) data from DONIMO v2 and conducts an improved calculation of tropospheric air mass factors (AMFs) and VCDs (i.e., VCD = SCD / AMF) (Boersma et al., 2011). Key features of the POMINO algorithm include explicit representation of aerosol scattering and absorption (by combining aerosol data from daily nested GEOS-Chem (at 0.3125°

long. × 0.25° lat.) simulations and monthly MODIS/Aqua aerosol optical depth (AOD) data), explicit representation of the angular dependence of surface reflection, high-resolution $NO_2$ profiles from GEOS-Chem (at 0.3125° long. × 0.25° lat.), consistent retrievals of clouds (a prerequisite for the $NO_2$ retrieval) and $NO_2$, and use of a parallelized, LIDORT-driven AMFv6 package. POMINO $NO_2$ VCDs are consistent with ground-based MAX-DOAS data (Liu et al., 2019).

To better relate $NO_x$ emissions to $NO_2$ VCDs at the 0.05°×0.05° resolution, we only employ the $NO_2$ data in summer (June, July, and August), in which season the lifetimes of $NO_2$ are the shortest (a few hours). We combine data over 2012–2015 to increase the sample size. The change in $NO_2$ VCD from June to August is relatively small, reducing the effect of intra-seasonal variability when deriving $NO_x$ emissions from summer mean $NO_2$ VCDs. We screen out the 30 outer pixels with VZA larger than 30° (cross-track

width larger than 36 km) that greatly smear the spatial gradient of $NO_2$, pixels with cloud radiance fraction exceeding 50%, and pixels with AOD larger than 3 (i.e., when the aerosol data used in the $NO_2$ retrieval are unreliable and the $NO_2$ retrieval is subject to an excessive error) (Lin et al., 2014; Lin et al., 2015b; Liu et al., 2018a). We also exclude data with raw anomaly problems (http://projects.knmi.nl/omi/research/product/rowanomaly-background.php). After data screening, we

obtain valid data from 22,007 pixels. We then convert the pixel-specific Level-2 data to the 0.05°×0.05° grid.

To convert from the satellite pixels to the 0.05°×0.05° grid cells, we use an oversampling method that employ satellite data on multiple days to enhance the horizontal resolution (Zhang et al., 2014). For each 0.05°×0.05° grid cell, we average all pixels covering the grid cell from all valid days, using area-based

weighting. The oversampling approach takes advantage of the fact that the exact location of the OMI footprint slightly changes from one day to another, so does the exact location of the footprint of an OMI

pixel at a given VZA. Thus, sampling from multiple days increases the horizontal resolution of data. Our oversampling approach is different from previous studies, which filled a grid cell with data from pixels within a certain distance (e.g., 30 km) and would result in spatial smoothing (Fioletov et al., 2011;Krotkov et al., 2016;Sun et al., 2018).

For the purpose of emission estimating, we assume that the error of VCD at a satellite pixel ($\sigma_p$) contains an absolute error of half of the mean VCD over the domain (i.e., $1.9 \times 10^{-15}$ molecules cm$^{-2}$) and a relative error of 30% (Lin et al., 2010; Boersma et al., 2011; Beirle et al., 2011; Lin et al., 2015a). We further add in quadrature an additional error ($\sigma_g$) when a satellite pixel is projected to the grid cells at a finer resolution; this error is important in the urban-rural fringe zone. For a given grid cell, $\sigma_g$ is set to be 50% of the
standard deviation of VCDs at its eight surrounding grid cells. Sampling over multiple days reduces the random error by a factor of $s = \left( \sqrt{(1-c)/n} + c \right)$, where c represents the fraction of systematic error (assumed to be 50%) and $n$ the number of days with valid data (Eskes et al., 2003; Miyazaki et al., 2012). Thus, the total error for the temporally averaged VCD at a given grid cell is $\sigma_s = \sqrt{(\sigma_p^2 + \sigma_g^2)} \cdot s$.

### 2.3. The PHLET model simulation

We construct the PHLET model on the 0.05°×0.05° grid to interpret the relationship between local net source (i.e., emission – loss), horizontal transport, and VCDs of NO$_2$ in a 2-D gridded space (Eq. 1) in the sense of long-time average. PHLET simulates the horizontal transport of NO$_x$ through a time averaged advection process and an "effective" diffusion process, which represents the residual from the temporally averaged advection. The vertical distribution is simplified as in Sect. 2.3.2. The loss process of NO$_x$ is
represented by the lifetime.

#### 2.3.1   Governing equation of PHLET

PHLET is an equilibrium model for the local net source, VCDs and horizontal transport of NO$_2$ at each grid cell. Equation 1 presents the governing equation in PHLET:

$$\frac{\partial C(x,y)}{\partial t} = r \cdot L(x,y) - \mathbf{V}(x,y) \cdot \nabla C(x,y) + \nabla \cdot \left(\mathbf{K}(x,y) \cdot \nabla C(x,y)\right) = 0 \qquad (1)$$

$C(x,y)$ represents the tropospheric $NO_2$ VCD (in molecules $cm^{-2}$) due to sources over the YRD. The value of $C(x,y)$ gives the distribution of $NO_2$ VCDs at equilibrium ($\frac{\partial C(x,y)}{\partial t} = 0$ for every x and y). The discrete form of PHLET is set on the $0.05° \times 0.05°$ grid. The simulated VCDs will be applied with the

SCM and compared to the gridded OMI data (after the contribution of horizontally homogeneous regional background is subtracted from the OMI data, see Sect. 2.3.2).

We assume a steady state of $NO_2$ in PHLET, although $NO_2$ observed by the satellite instrument may be in a transient state. We assign an error of 15% to simulated $C(x,y)$ to account for the possible range of $NO_2$ variability at the overpass time of the instrument. Also, combining data from multiple years to derive

an averaged $NO_2$ distribution for simulation (rather than conducting the simulations for individual years and months) leads to an additional uncertainty, which is set to be 10% based on a comparison between the emissions estimated from multiple years together (here) and the average of emissions estimated from individual years (in a sensitivity test).

$L(x,y)$ represents the local net source term (in molecules $cm^{-2}$ $s^{-1}$, equivalent to $2.63 \times 10^{-12}$ kg $km^{-2}$ $h^{-1}$),

which combines the effects of ground emissions (anthropogenic + soil + biomass burning; see discussion in Sect. 2.3.2), deposition, and chemistry of $NO_x$. At equilibrium, the domain average of modeled $L(x,y)$ reaches zero, because there are no horizontal fluxes into or out of the domain boundaries. $L(x,y)$ can be separated into an emission term and a loss term:

$$L(x,y) = E(x,y) - \frac{C(x,y)}{r \cdot \tau(x,y)} \qquad (2)$$

where $E(x,y)$ denotes the gridded emissions of $NO_x$, and $\tau(x,y)$ the lifetimes associated with deposition and chemical loss. $r$ represents the ratio of $NO_2$ over $NO_x$ concentration. The daytime $NO_x$ chemical system reaches equilibrium rapidly and $r$ varies little (Beirle et al., 2011;Valin et al., 2013). We set $r$ to be 0.76 with an uncertainty of 10% (Seinfeld et al., 2006; Beirle et al., 2011).

$\boldsymbol{V}(x,y) \cdot \nabla C(x,y)$ represents the time averaged advection term. $\boldsymbol{V}(x,y)$ denotes the mean wind vector (in m s$^{-1}$) averaged over summer 2012–2015. The wind data are taken from the European Centre for Medium-range Weather Forecasts (ECMWF) ERA5 dataset (see details in Sect. 2.3.3).

$\nabla \cdot (\mathbf{K}(x,y) \cdot \nabla C(x,y))$ represents the diffusion term, where $\mathbf{K}(x,y)$ denotes the "effective" diffusion coefficient tensor (in m$^2$ s$^{-1}$). The diffusion term accounts for transport by the residual winds deviating from the temporally averaged wind vector $\boldsymbol{V}(x,y)$. Appendix A shows how to determine the diffusion coefficient tensor.

### 2.3.2  *Vertical shape and regional background of NO$_2$*

PHLET assumes a horizontally homogeneous vertical shape of NO$_2$ concentrations, and that NO$_2$ is concentrated near the surface (Beirle et al., 2011). The assumption is implicitly used in many previous studies for polluted areas (Beirle et al., 2011; Liu et al., 2016; Liu et al., 2017). The corresponding uncertainty in the modeled NO$_2$ VCDs is set to 15% (Boersma et al., 2011; Lin et al., 2014).

Lightning emissions, biomass burning emissions, aircraft emissions, transport from neighboring regions, and convection can lead to NO$_2$ at higher altitudes over the YRD area. However, the amount of NO$_2$ aloft is much smaller than near-ground NO$_2$ due to large ground sources (Lin, 2012). Thus, we regard NO$_2$ aloft as the regional background, and do not include it in Eq. 1. Also, for near-ground NO$_2$ over the YRD area, the contribution of downward vertical transport is negligible compared to the contribution of ground sources. Aircraft emissions contribute little to the total ground source, because 78% of aircraft emissions occur at the high altitudes (9–12 km) (Ma and Xiuji, 2000). Therefore, PHLET only accounts for near-ground NO$_2$ from ground soil, biomass burning and anthropogenic sources (energy, industry, transportation, and residential).

To ensure the consistency between PHLET and OMI NO$_2$ data, we assume the background value to be half of the minimum OMI NO$_2$ VCD among all grid cells (i.e., $0.54 \times 10^{15}$ molecules cm$^{-2}$), and then

subtract the background value from the gridded OMI data when comparing with PHLET simulations. The corresponding uncertainty in the modeled $NO_2$ VCDs is set as 5%.

### 2.3.3 Initial conditions, lateral boundary conditions, and wind data input

To run PHLET, the $NO_2$ VCDs at the domain edges, as the lateral boundary conditions (LBCs), are set as the corresponding OMI $NO_2$ VCDs. For initial conditions, the VCD and the local net source at each grid cell inside the domain boundaries are set as zero. The horizontal distributions of modeled $NO_2$ VCDs and local net sources at equilibrium do not depend on the initial conditions.

For horizontal transport, we use 3-hourly wind fields from the ECMWF ERA5 dataset (https://confluence.ecmwf.int//display/CKB/ERA5+data+documentation; last access: 2018/7/2). The resolution of raw ERA5 data is 0.28125° on the reduced Gaussian grid, which is regridded to 0.05°×0.05° by using the online program offered by ECMWF (see Fig. A). We adopt the mean wind field of the lowest 14 vertical levels (out of 157 levels in total); these 14 levels represent the altitudes from surface to about 500 m (Beirle et al., 2011; Hersbach and Dee, 2016). Over the study period, the prevailing wind is northwesterly, and the wind speed is small over land (Fig. A). For both zonal and meridional wind speeds, the uncertainty in the average wind speed is set to be 10%, which is similar to the temporal standard deviation of the wind speed and may partly account for the fact that lower-resolution wind data are used. We assess the model errors introduced by the uncertainties in the wind field and effective diffusion coefficients by Monte Carlo simulations in which the wind speeds are changed according to their uncertainties. The resulting relative uncertainty in the modeled $NO_2$ VCDs is about 20%.

### 2.3.4 Application of SCM

Re-mapping of PHLET simulated $NO_2$ VCDs in accordance to satellite pixels is important. Given the size of OMI pixels, the OMI $NO_2$ data smooth to some extent the actual horizontal distribution of $NO_2$. To ensure consistent spatial sampling between PHLET and OMI data, for each day we project the PHLET modelled $NO_2$ VCD data (in the original 0.05°×0.05° grid) to the satellite pixels to mimic how OMI "sees" the ground, remove the pixels with invalid OMI data, and then project the model data back to the

0.05°×0.05° grid. The last two procedures are the same as done for OMI data. The whole process of grid conversion is done through the SCM approach (Appendix B). Although PHLET simulates summer average NO₂ VCDs (rather than daily values), we repeat the grid conversion process for as many days as there are valid OMI data.

5            *2.3.5   Summary of model errors*

The model error $\sigma_m$ is set to be the sum in quadrature of errors contributed by the above mentioned steady state assumption (15%), the time averaging over multiple years and months (10%), the assumption of horizontally constant vertical shape of NO₂ (15%), the NO₂/NO$_x$ ratio (10%), the treatment of background NO₂ concentration (5%), and the error in the wind data and the calculation of effective diffusion coefficients (20%).

## 2.4.  PHLET-A: The adjoint model of PHLET

We construct the PHLET-A adjoint model to obtain an optimized horizontal distribution of the local net source term (*L* in Eq. 1) under the given OMI NO₂ VCDs, wind field, and other parameters. PHLET-A accounts for the complex nonlinear effects of 2-dimensional transport and loss processes.

15  We first define a scalar cost function (Eq. 3) to quantify the difference between OMI NO₂ VCDs and PHLET simulated (and SCM applied) NO₂ VCDs.

$$J = (\boldsymbol{C}^{OMI} - \boldsymbol{C}^{PHLET})^T \mathbf{S_o}^{-1} (\boldsymbol{C}^{OMI} - \boldsymbol{C}^{PHLET}) \tag{3}$$

Because PHLET does not require a priori knowledge about the local net source, the cost function does not include the a priori term either. The vector $\boldsymbol{C}$ denotes gridded NO₂ VCDs. $\mathbf{S_o}$ denotes the observational error covariance matrix consisting of a satellite data error covariance matrix ($\mathbf{S_s}$) and a PHLET model error covariance matrix ($\mathbf{S_m}$):

$$\mathbf{S_o} = \mathbf{S_s} + \mathbf{S_m} \tag{4}$$

For simplicity and following previous studies (Keiya and Itsushi, 2006; Cao et al., 2018), both $\mathbf{S_s}$ and $\mathbf{S_m}$ are assumed to be diagonal, with the diagonal elements set to be $\sigma_f^2$ and $\sigma_m^2$, respectively. Grid cells nearby may share the same pixels, although the area-based weights would be different. This means that nearby grid cells may not be fully independent, leading to a weakness of the diagonal assumption here. The associated uncertainty is partly accounted for by an error term based on the variability of NO$_2$ VCDs (i.e., 50% of the standard deviation across the surrounding grid cells; see Sect. 2.2).

We then derive PHLET-A and its initial and lateral boundary conditions by applying Lagrange identity and integrating by parts (Marchuk, 1994; Sandu et al., 2005; Martien et al., 2006; Hakami et al., 2007):

$$\frac{\partial \lambda(x,y)}{\partial t} + \nabla(\mathbf{V}(x,y) \cdot \lambda(x,y)) + \nabla \cdot (\mathbf{K}(x,y) \cdot \nabla \lambda(x,y)) = 0 \tag{5}$$

$$\lambda(x,y)|_{t=T} = \frac{\delta J}{\delta C(x,y)}\Big|_{t=T} \tag{6}$$

$$\lambda(x,y)|_{boundary} = 0 \tag{7}$$

As shown in Eq. 5, PHLET-A represents the sensitivity of cost function ($J$) to local net source ($\mathbf{L}$) where $\lambda$ stand for the adjoint variable (Marchuk, 1994; Sandu et al., 2005). $T$ stands for the time when the domain-wide NO$_2$ VCDs come to equilibrium, i.e., the start time of the adjoint simulation. By discrete adjoint sensitivity analysis, the gradient of $J$ to $\mathbf{L}$ is obtained:

$$\frac{\delta J}{\delta L_{i,j}} = r \cdot \Delta x \cdot \Delta y \cdot \Delta t \cdot \sum_k \lambda_{i,j,k} \tag{8}$$

where the indices $i$, $j$ and $k$ denote zonal, meridional, and time, respectively. The gradient is then used in an iterative optimization (shown by the blue arrows in Fig. 1) to minimize the cost function $J$, i.e., to minimize the weighted difference between model simulated and OMI NO2.

The numerical solution to obtain an optimized $\mathbf{L}$ that minimizes $J$ is as follows. Given a starting point of $\mathbf{L}$, we derive a search direction by the Broyden-Fletcher-Goldfarb-Shanno (BFGS) method (Li and

Fukushima, 2001; Bousserez et al., 2015). Then, by practicing backtracking line search based on the Armijo–Goldstein condition (Armijo, 1966), we obtain a revised $L$ for the next iteration. The numerical calculation is done through FEniCS. It takes 50 iterations of PHLET and PHLET-A runs before the convergence is reached, according to the rate of reduction in $J$. The value of $J$ is reduced from an initial

value of 6585.2 to a stabilized value of 73.6 (Fig. 2).

The uncertainty of the optimized $L$ is given by the Hessian of the cost function, which is approximated by the BFGS method (Brasseur and Jacob, 2017):

$$S = 2 \cdot (\nabla_L^2 J)^{-1} \tag{9}$$

## 2.5.   Deriving emission and loss from the local net source term

The optimized local net source term combines the contributions of emission and loss (chemical loss + deposition). We further separate emission from loss by assuming a fixed formula within our small study domain for the nonlinear relationship between lifetimes and VCDs of $NO_2$.

In the summertime daytime, the dominant sink of $NO_x$ is reactions with the radicals to produce nitric acid and organic nitrogen species. The $NO_x$ chemistry quickly reaches a steady state under high solar radiation

and air temperature in the early afternoon (Murphy et al., 2006; Valin et al., 2013) when OMI passes over the YRD. The chemical lifetime of $NO_x$ depends on the concentrations of $NO_x$ and non-methane volatile organic compounds (NMVOC), radiation, temperature, and other factors. Within our small study domain, we assume the net effect of all factors except $NO_x$ concentrations to be spatially homogeneous. As such, the chemical lifetime of $NO_x$ at steady state is a sole function of $NO_x$ concentration (and thus $NO_2$ VCD,

given the constant $NO_2/NO_x$ ratio). Appendix C shows in detail how to deduce the chemical lifetime of $NO_x$ from $NO_2$ VCD, to account for the effect of dry deposition, to separate emission and lifetime from the local net source term, and to quantify the errors involved.

## 2.6. Uncertainty estimate for top-down emissions

For a particular grid cell, the derived emission is affected by the error involved in the estimate of $L$ (embedded in satellite data and model simulations) and the error in the separation of emission and lifetime from $L$. The satellite data error $\sigma_s$ is analyzed in Sect. 2.2. The model related error $\sigma_m$ is analyzed in Sect. 2.3. The error of $L$, $\sigma_L$, is connected with $\sigma_s$ and $\sigma_m$ through the adjoint simulation and is given by Hessian of the cost function $J$ (Sect. 2.4).

The error involved in the separation of emission and lifetime, $\sigma_f$ , is contributed by the assumption on the NO$_2$/NO$_x$ ratio r (Sect. 2.3.1), the simplified treatment of deposition and chemical processes of NO$_x$ (Appendix C), and the assumed relationship between lifetimes and VCDs (Appendix C). $\sigma_f$ is estimated by data fitting of $L$ with different fitting parameters (Appendix C).

Thus, the error in emission, $\sigma_e$, is equal to the sum in quadrature of $\sigma_L$ and $\sigma_f$, i.e., $\sigma_e = \sqrt{\sigma_L{}^2 + \sigma_f{}^2}$. The error in the lifetime is derived from the errors in NO$_x$ loss (estimated in Appendix C) and NO$_2$ VCDs, according to the common manner of error synthesis.

## 3. High-resolution spatial distributions of NO$_2$ VCDs, local net sources, and lifetimes over the YRD

Figure 3a shows the number of days with valid OMI data in summer 2012–2015 over the YRD area on the 0.05°×0.05° grid. The number of days varies from about 12 to 101 (48 on average). There is a "band" pattern in the spatial distribution, due to the difference in the number of satellite orbits covering each grid cell (not shown). This band pattern is not obvious in the distribution of OMI NO$_2$ VCDs (Fig. 3b), suggesting that the temporally averaged VCD values are less sensitive to the number of days (12 or more) used for temporal averaging. There are fewer valid data in severely polluted locations. The effect of sampling size on the uncertainty in OMI NO$_2$ is accounted for in our study (Sect. 2.2).

Figure 3b shows the gridded horizontal distribution of OMI $NO_2$ VCDs. The background value ($0.54\times10^{15}$ molecules $cm^{-2}$) has not been removed. $NO_2$ VCDs are high over the major urban centers along the Yangtze River and the coastal line, especially Shanghai, Nanjing (Capital of Jiangsu Province), Hangzhou (Capital of Zhejiang Province), and the Ningbo-Zhoushan area (with intensive maritime shipping activities). The maximum VCD value exceeds $16\times10^{15}$ molecules $cm^{-2}$ in north Shanghai. $NO_2$ VCDs are larger than $1\times10^{15}$ molecules $cm^{-2}$ at all grid cells, reflecting the influence of local anthropogenic sources and/or pollution transported from nearby cities (Cui et al., 2016). $NO_2$ VCDs are lower than $5\times10^{15}$ molecules $cm^{-2}$ along the boundaries of our study domain.

Across the grid cells, the absolute errors in OMI $NO_2$ VCDs are about $1.6$–$4.9\times10^{15}$ molecules $cm^{-2}$ (Fig. 4a), and the relative errors are about $30\%$–$157\%$ (Fig. 4b). In general, the grid cells with larger $NO_2$ VCDs have larger absolute errors but smaller relative errors. Over the eastern sea and the southwestern corner of the domain, $NO_2$ VCDs are relatively small (Fig. 3b), thus their absolute errors are small (Fig. 4a), but their relative errors are very large (Fig. 4b).

Figure 4c shows the spatial distribution of the local net source $L$ (emission – loss). A positive (negative) value of L indicates that the emission is larger (smaller) than the loss. The values of $L$ are the greatest ($11.7$ kg $km^{-2}$ $h^{-1}$) over the major urban areas with high $NO_2$ VCDs, and are low ($< -1.0$ kg $km^{-2}$ $h^{-1}$) in many areas with low $NO_2$ loadings. The values of $L$ are the lowest ($(-6.9)$– $(-2.0)$ kg $km^{-2}$ $h^{-1}$) at places in the urban-rural fringe zones with $NO_2$ hotspots nearby; this feature reflects that $NO_2$ is transported from the urban centers and destroyed in the fringe zones. The absolute errors of $L$ vary from $0.6$ to $4.5$ kg $km^{-2}$ $h^{-1}$. The spatial correlation between the absolute errors of $L$ (Fig. 4c) and those of the $NO_2$ VCDs (Fig. 4a) is about $0.5$. The absolute errors of $L$ are notable in the urban-rural fringe zones where $L$ is small but $NO_2$ VCD is high, because the deviation of $L$ at these areas is very sensitive to errors in the assumed transport and loss process.

Figure 3d shows the derived lifetimes of $NO_2$ on the $0.05°\times0.05°$ grid. The lifetimes range from $0.6$ to $3.3$ h across the study domain, with an average of $2.0$ h. The lifetimes are about $0.6$ h at grid cells with

NO$_2$ VCDs of about $1.6\times10^{15}$ molecules cm$^{-2}$, increasing to 0.8 h at grid cells with the lowest VCDs (around $1.0\times10^{15}$ molecules cm$^{-2}$), and exceeding 3 h at many polluted grid cells, i.e., the urban centers. The nonlinear dependence of lifetimes on VCDs is expected from our inversion method (Appendix C). Appendix C further shows the chemical lifetimes to be 0.6–3.8 h and the deposition lifetimes to be constantly at 30.4 h across all grid cells.

Figure 4d shows that the absolute uncertainties in the lifetimes are greater than 1.0 h at the NO$_2$ hotspot locations, between 0.6 and 1.0 h over the eastern sea and the southwest of the study domain, and about 0.4 h at many other locations. The lifetimes at high-NO$_2$ locations (up to 3.3 h) are consistent with previous studies, e.g., one single value of 4.7±1.4 h for Shanghai in summer 2005–2013 by (Liu et al., 2016). The short lifetimes at low-NO$_2$ locations over the eastern sea may be underestimated, due to the assumption that non-NO$_2$ factors (especially NMVOC) are spatially homogeneous within the study domain (Appendix C). In particular, the concentrations of NMVOC over the eastern sea may be overestimated by this assumption, based on the OMI formaldehyde data (De Smedt et al., 2015).

## 4.  High-resolution spatial distribution of NO$_x$ emissions over the YRD

### 4.1.  Spatial distribution of emissions

Figure 5a shows the derived horizontal distribution of summer 2012–2015 average NO$_x$ emissions on the 0.05°×0.05° grid. The emissions include the contributions of ground anthropogenic (energy, industry, transportation, and residential), soil and biomass-burning sources. As discussed in Sect. 4.3, soil emissions contribute little (0.5%) to the total emissions over the study domain, and biomass burning contributes about 5.1%.

Figure 5a shows that NO$_x$ emissions vary from 0 to 15.3 kg km$^{-2}$ h$^{-1}$ across the grid cells. The highest emission value occurs in north Shanghai, close to Wusongkou where Shanghai Port is located, which has become the largest container terminal in the world in 2010 (Fu et al., 2012). High emission values also occur at places along the Yangtze River and the coastal line. Along the Yangtze River, the highest

emission value occurs in Nanjing City. Along the coastal line, there is an emission hotspot in the Ningbo-Zhoushan area. The general spatial distribution of $NO_x$ emissions is consistent with that of OMI $NO_2$ VCDs (correlation = 0.69), reflecting the short lifetimes of $NO_x$ and thus a modest effect of horizontal transport. Nonetheless, emissions are much more concentrated at a few sparse locations than $NO_2$ VCDs

are, and many locations near the emission hotspots have very low emissions but relatively large $NO_2$ VCDs, suggesting that the effect of horizontal transport cannot be ignored at such a high resolution.

Figure 5b shows the absolute errors of $NO_x$ emissions at individual grid cells. The emission errors vary from 0.7 to 4.5 kg km$^{-2}$ h$^{-1}$ across all grid cells. The largest uncertainty occurs in north Shanghai, corresponding to the highest VCD (Fig. 3b) and emission (Fig. 5a) values. The spatial pattern of absolute

emission errors (Fig. 5b) is closer to the pattern of $NO_2$ VCDs (Fig. 3b, correlation = 0.51) than to the pattern of emissions (Fig. 5a, correlation = 0.33). There are more hotspots in the distribution of emission errors than in the distributions of VCDs and emissions, because the emission errors can be high at locations with low VCDs and emissions. The spatial pattern of emission errors is consistent with that of $L$ errors (Fig. 4c, correlation = 1.0), indicating that the errors in deriving emissions from the local net

sources are rather homogenous. Figure 5c further shows that the relative errors of emissions are high (> 100%) over low-emission locations but much lower over emission hotspots.

The scatter plot in Fig. 5d shows the relationship between absolute emission errors and emissions at individual grid cells. The relationship is highly nonlinear, and there is large data spread where the emissions are low. The data spread tends to be smaller when emissions exceed 5 kg km$^{-2}$ h$^{-1}$. The emission

errors tend to decrease as emissions increase until about 5 kg km$^{-2}$ h$^{-1}$, after which the emission errors tend to increase with the increasing emissions. The data points in Fig. 5d are colored to indicate the different ranges of VCDs, and they show that grid cells with higher $NO_2$ VCDs have larger emission errors and smaller data spread.

## 4.2.  Comparison between our top-down emissions and spatial proxies

This section compares our $NO_x$ emission dataset (Fig. 6b) with several spatial proxies widely used in bottom-up inventories, including nighttime light brightness (Fig. 6c), population density (Fig. 6d), road network (Fig. 6e), ship route density (Fig. 6f), power plant locations (Fig. 6g), and a satellite photo from Google Earth based on Landsat measurement that indicates the extent of land use (Fig. 6i). These proxies broadly represent the intensity of human activities and are highly related to $NO_x$ emissions (Geng et al., 2017).

Figure 6c shows the spatial distribution of nighttime light brightness in 2012. The data are taken from Version 4 DMSP-OLS Nighttime Lights Time Series at a horizontal resolution of 0.5'×0.5' (https://www.ngdc.noaa.gov/eog/dmsp/downloadV4composites.html; last access: 2018/08/19). The brightness is represented digitally from 0 to 63 bits. The nighttime light reflects the intensity of household activity, commercial activity, and resource consumption (Elvidge et al., 2013). When regridded to 0.05°×0.05°, the spatial correlation between nighttime light brightness and $NO_x$ emissions is about 0.61 over land.

Figure 6d shows the population density data, which are taken from the Gridded Population of the World v4 (GPWv4) at a horizontal resolution of 0.1°×0.1° (http://sedac.ciesin.columbia.edu/data/collection/gpw-v4/sets/browse; last access: 2018/08/19) (Center for International Earth Science Information Network - CIESIN - Columbia University, 2016). This dataset provides population density data for every five years (2000, 2005, 2010, 2015, etc.). Data in 2012, 2013 and 2014 are estimated by fitting a natural spline to the 2000, 2005, 2010, and 2015 values. The population density varies greatly from the urban centers to the countryside. In north Shanghai, the population density exceeds $3.5×10^3$ km$^{-2}$. The $NO_x$ emission hotspots match the population hotspots, and the lowest-emission locations have little population. When regridded to 0.05°×0.05°, the spatial correlation between population densities and $NO_x$ emissions is 0.50 over land.

Figure 6e shows the OpenStreetMap road network data (http://download.geofabrik.de; last access: 2018/6/27). The network includes both highways and local roads. In the southern areas (between 29°N and 31°N), the spatial distribution of $NO_x$ emissions largely coincides with the road network. The spatial coincidence is less obvious in the north because of the influence of non-mobile sources. $NO_x$ emissions are notable along the three major national highways connecting Jinhua City (one of the largest hubs of light industry products in China), Hangzhou City, and Ningbo City. $NO_x$ emissions are also identifiable along the national highway from Hangzhou City to Huangshan City. Pairs of $NO_x$ emissions and traffic hubs are located to the west of the Taihu Lake and in the urban centers. These results suggest the capability of our emission dataset in capturing the contribution of traffic sources.

Figure 6f shows the mean density of marine shipping routes over the eastern sea in 2016 (www.marinetraffic.com; last access: 2019/6/27). Over the northern parts of the eastern sea, the route density map shows certain north-south and northwest-southeast lines. High route densities are also evident close to the ports. These features are consistent with the distributions of $NO_2$ VCDs (Fig. 6a) and $NO_x$ emissions (Fig. 6b, same as Fig. 5a).

The filled circles in Fig. 6g show the locations of coal-fired power plants in 2016 from Carbon Brief (www.carbonbrief.org; last access: 2019/6/27). The radius of a circle denotes the power generation capacity. Figure 6h further shows the GPED v1.0 bottom-up $NO_x$ emissions for power plants on a 0.1°×0.1° grid in 2016. Coal-fired power plants in the YRD are normally near the urban centers, traffic lines or other sources. Our top-down $NO_x$ emission map shows large emission values near the power plants (Fig. 6b), although it cannot isolate the sole contribution of power plants. At the GPED power plant locations, the correlation between our and GPED emissions reaches 0.26, due to the influence by non-power plant sources; note that the correlation between GPED emissions and POMINO $NO_2$ VCDs are only about 0.21.

Figure 6i shows a satellite photo taken in 2018 from Google Earth (earth.google.com; last access: 2019/7/4). The grey areas in the photo represent developed lands and the dark green areas indicate un-

developed places. The majority of lands over the YRD have been developed. Although the lands over the southwest are less influenced by humans than the areas like Shanghai are, many places of the southwest have been developed as cities, towns and roads. This explains the spotted emission sources (Fig. 6b) retrieved from the satellite $NO_2$ VCDs.

**4.3. Comparison between our emission dataset and other inventories**

This section compares our emission data to several inventories for the region, including the MEIC bottom-up anthropogenic inventory in summer 2012–2015 (www.meicmodel.org; last access: 2018/7/2) (Liu et al., 2015;Zheng et al., 2014), the MarcoPolo (bottom-up + top-down hybrid) anthropogenic inventory in summer 2014 (www.marcopolo-panda.eu/products/toolbox/emission-data/; last access: 2019/5/4) (Hooyberghs et al., 2016), and the DECSO v5.1qa top-down emissions in summer 2012–2015 (www.globemission.eu/region_asia/datapage.php; last access: 2018/11/14) (Mijling et al., 2013; Ding et al., 2017a). MEIC and DESCO emission data are available at the 0.25°×0.25° resolution, and MarcoPolo are at 0.01°×0.01°. When compared with our emissions, these emission data are regridded to 0.05°×0.05°.

Our emission data and the DECSO inventory are top-down estimates and include the contributions of soil and biomass-burning sources. Thus, we estimate soil and biomass burning emissions from independent sources, and then subtract these emissions from our and DECSO emission datasets. Soil emissions are calculated by the nested GEOS-Chem (Fig. 7c), with the uncertainties assumed to be within 50% (J. Yienger and Ii Levy, 1995; Wang et al., 1998). Biomass burning emissions (Fig. 7b) are taken from the Global Fire Emissions Database (GFED4; www.globalfiredata.org/data.html; last access: 2019/7/10) (Giglio et al., 2013), with the uncertainties estimated to be within 10% over the YRD (Giglio et al., 2009; Giglio et al., 2013). Summed over the study domain, the soil sources contribute about 0.5% of our emissions while biomass burning contribute about 5.1%. Figure 7a shows the resulting "anthropogenic" portion of our emissions.

Compared with MEIC (Fig. 7d) and DECSO (Fig. 7e), our high-resolution anthropogenic emission dataset (Fig. 7a) provides much more detailed spatial information. Our dataset identifies the emission hotspots

and their contrast with nearby low-emission areas (e.g., in the urban-rural fringe zones) better than MEIC and DECSO do. The contribution of mobile sources along the road network is clearer in our dataset. Our emission data contain sources over the nearby sea (i.e., from shipping), along the coastal line, and in the southwest of the domain, which are not included in MEIC. Compared with DECSO, our dataset suggests

higher emissions on the northern parts of the sea, which may be due to our underestimate of $NO_x$ lifetimes (Sect. 3) and/or errors in the DECSO estimate.

MarcoPolo emissions (regridded to 0.05°×0.05°, Fig. 7f) show more detailed spatial information than our emission dataset (Fig. 7a) does. This is because our top-down estimate is limited by the intrinsic resolution of $NO_2$ VCDs, i.e., our oversampling approach does not fully compensate for the large sizes of OMI

pixels. Therefore, the large spatial gradient of $NO_x$ emissions is smoothed to some extent in our dataset. On the other hand, the domain that MarcoPolo covers $(118.135°E, 29.635°N - 122.125°E, 32.625°N)$ is much smaller than ours, and emissions of 9 cities (including Zhou Shan, Ningbo, Nantong, Hangzhou, Huai'an, Yancheng, Yangzhou, Taizhou, Shaoxing) and marine shipping emissions are not included in MarcoPolo.

At 0.05°×0.05°, the spatial correlations between our $NO_x$ emissions and spatial proxies are 0.61 for nighttime light brightness and 0.50 for population density (Sect. 4.2). These values can be compared to the respective results for MarcoPolo on the 0.05°×0.05° grid (0.35 and 0.55, respectively). When regridded to 0.25°×0.25°, the correlations between our emissions and these spatial proxies become higher: 0.70 for nighttime light brightness and 0.69 for population density. The weaker correlation at a higher

resolution reflects that as the spatial resolution gets finer, the chance that $NO_x$ emissions are collocated with population or nighttime light becomes smaller (Zheng et al., 2017), because of the influences of $NO_x$-emitting factories, power plants, and mobile sources. By comparison, the correlations between MEIC and these proxies on the 0.25°×0.25° grid are 0.80 for nighttime light and 0.81 for population density. The respective correlation values for DECSO are 0.66 and 0.46. The lower correlation values for

our dataset and DESCO than for MEIC partly reflect that top-down emissions better account for the

influences of land transportation, which are spatially not tied closely to nighttime light and population at this resolution.

Figure 8 compares city-level emissions between our and other inventories. A total of 18 cities within the domain are selected, and for each city the $NO_x$ emissions are summed over the grid cells within the municipal administrative boundaries. All inventories are gridded at 0.05°×0.05° for this purpose. The MarcoPolo inventory does not include emissions for several cities, thus the respective color bars are missing from Fig. 8. Among the cities, emission values differ from -5.8% to +67.5% between our emissions and the mean values of all four inventories (ours, DECSO, MEIC and MarcoPolo (if available)). For most cities, our emissions are consistent with at least one of the other three inventories, often the DESCO top-down inventory, after accounting for errors in our emission estimate. In Yancheng, Huai'an and Ningbo, our emission values are higher than the averages of ours, DECSO and MEIC by 67.5%, 57.5% and 34.6%, respectively. Ningbo (around 29.8°N, 121.5°E) is a coastal city with many isles and marine ports, as identifiable on the nighttime light map (Fig. 6c). The marine ports in the Ningbo-Zhoushan area contribute about 10% of the total shipping emissions in China (Endresen et al., 2003; Fu et al., 2017). Our dataset and DECSO account for emissions from marine shipping and ports, whereas MEIC does not.

## 5. Test of our top-down emission derivation method by using GEOS-Chem simulated $NO_2$ data

This section further presents an test to estimate the reliability of our emission derivation method, by examining to what extent the method can re-produce the emissions used in a nested GEOS-Chem simulation. Specifically, we use the nested GEOS-Chem v9-02 (Yan et al., 2016; Liu et al., 2018b; Ni et al., 2018) to simulate the $NO_2$ VCDs in the early afternoon (around the overpass time of OMI) in summer 2014 on the 0.3125° longitude ×0.25° latitude grid. The simulated $NO_2$ data are shown in Fig. 9a and the emission inputs are shown in Fig. 9b. Next, we convert the GEOS-Chem $NO_2$ VCDs into the 0.05°×0.05° grid, and parameterize PHLET with the wind field adopted by GEOS-Chem, following the procedures in Sect. 2.3. Then we use PHLET, PHLET-A, and the lifetime-emission separation method to estimate the

NO$_x$ emissions. Finally, we compare the derived emissions (re-mapped to the 0.3125°×0.25° grid) to those used in GEOS-Chem.

Figure 9c and 9d shows the horizontal distributions of our "anthropogenic" emissions and emission errors, respectively. The contribution of soil and biomass burning emissions (as simulated by GEOS-Chem) are

subtracted from the dataset. Figure 9e shows the differences between the derived anthropogenic emissions and those used in GEOS-Chem. The emission difference at each grid cell varies from−3.0 to 5. 4 kg km$^{-2}$ h$^{-1}$, which is attributed to the limitation of our inversion method. The domain average difference is 0.28 kg km$^{-2}$ h$^{-1}$, or 18% of GEOS-Chem emissions. The scatter plot in Fig. 9f suggests excellent consistency between the derived and the GEOS-Chem emissions, with a linear regression slope of 1.06 and correlation

of 0.94. The emission differences for most grid cells are within the uncertainties of the derived emissions (shaded area).

## 6.    Concluding remarks

This study presents a satellite-based top-down method to estimating NO$_x$ emissions over urban and surrounding areas at a high horizontal resolution. As a demonstration, the method is applied to the YRD

area at the 0.05°×0.05° resolution in summer 2012–2015, based on the POMINO NO$_2$ product. We construct a simplified, computationally efficient 2-D lifetime-emission-transport model (PHLET) and its adjoint model (PHLET-A) to, together with other procedures, facilitate the emission estimate. The reliability of our inversion method is supported by 1) a rigorous step-by-step derivation of models, assumptions, and parameters used, 2) a comprehensive uncertainty analysis, and 3) an test with GEOS-

Chem simulated NO$_2$ data. Our emission dataset in the YRD area on the 0.05°×0.05° grid shows fine-scale spatial information that is tied to nighttime light, population density, road network, maritime shipping, and land use indicated from a Google Earth photo. Our dataset reveals many fine-scale spatial characteristics not well represented or not included in lower-resolution inventories such as MEIC and DECSO. Although this study derives the averaged emissions over summer 2012–2015, calculations of

emissions at higher temporal resolutions (e.g., every 2 years) is possible to better capture the interannual variability and trends.

Our inversion method is useful for understanding how human activities have altered the atmospheric environment at fine resolutions. Many crucial human activities, such as urbanization, are conducted at very fine spatial scales. How the resulting emissions affect air quality, public health, and geo-health are still poorly understood due to lack of high-resolution emission data. This problem is particularly severe in the developing countries, because of their rapid paces of urbanization and great inadequacies in emission-related information such as economic statistics and emission factors. This poses a grand challenge for emission control and environmental management. Thus, our inversion method and resulting emission data offer useful independent high-resolution information to monitor the fine-scale emission sources, to improve the bottom-up inventory, to model the urban pollution chemistry and the effect of urbanization, and to conduct spatially targeted emission control.

Our inversion method also has a few shortcomings. The derived emissions do not separate the individual contributions of anthropogenic sectors (i.e., power plants, industry, transportation, and residential). The spatial resolution of the estimated emissions is limited by that of satellite VCD data, although a special oversampling technique has been used to help achieve the highest spatial resolution possible for emissions. The PHLET model is assumed to be 2-dimensional by simplifying the vertical distribution of $NO_2$ and not accounting for the spatial variability in the vertical shape, similar to previous studies. The adjoint model assumes the observational error covariance matrix to be diagonal, without fully considering the effect of correlations between individual grid cells. Also, we assume a spatially uniform relationship between $NO_2$ VCDs and $NO_2$ lifetimes, which may lead to an underestimate in the lifetimes at low-$NO_2$ locations over the eastern sea.

Our emission inversion method and models have a few important features enabling their global applications. PHLET and PHLET-A are written in the Python language, which can be readily used with low financial costs. The PHLET model offers computationally efficient simulations of the $NO_x$ chemistry,

deposition, and transport. At a low computational cost, our inversion method is able to account for the nonlinear relationship between $NO_x$ concentration, chemical loss, deposition, and transport. With the advent of TROPOMI and other satellite sensors with unprecedented spatial resolutions, our inversion approach can be applied to these measurements for continuous inference of emissions at finer and finer resolutions.

**Data availability**

Observational data are obtained from individual sources (see links in the text and acknowledgments). Model results are available upon request. Model codes are available on a collaborative basis.

**Author contributions**

JL conceived the research. HK, RZ and JL designed the research. HK and RZ performed the data processing, model development, and simulations. ML, HW, LC, RN, JW and YY contributed to data processing, model simulations, and data analyses. QZ provided MEIC data. HK and JL analyzed the results and wrote the paper with input from all authors.

**Competing interests**

The authors declare that they have no conflict of interest.

**Acknowledgements**

This research is supported by the National Natural Science Foundation of China (41775115) and the 973 program (2014CB441303).

**Appendix A. Solving the diffusion process**

The diffusion term can be simplified as follows:

$$\nabla \cdot \left( \mathbf{K}(x,y) \cdot \nabla C(x,y) \right) = \frac{\partial}{\partial x} \left( K_x \cdot \frac{\partial c}{\partial x} \right) + \frac{\partial}{\partial y} \left( K_y \cdot \frac{\partial c}{\partial y} \right) \qquad\qquad (A1)$$

$K_x$ and $K_y$ are the diffusion coefficients in the zonal and meridional directions, respectively.

We derive the diffusion coefficients based on a random walk assumption (Schirmacher, 2015):

$$K_{x\,or\,y} = \frac{1}{2} \overline{V_{x\,or\,y}'}^{2} t_0, \qquad\qquad (A4)$$

$\overline{V_{x\,or\,y}'}$ is the deviation of wind speed in the zonal or meridional direction. $t_0$ is 3 hours, the sampling interval of ERA5 wind data. Figure A shows the time averaged wind vector and the distribution of $K_x$ and $K_y$. The relative uncertainty in wind speed is assumed to be 10%, close to the temporal standard deviation of wind speed. The uncertainties of $K_x$ and $K_y$ are set to be 20%, about twice of the relative uncertainty in wind speed. The calculated $K_x$ ranges from 30397 $m^2 s^{-2}$ over land to 203783 $m^2 s^{-2}$

over sea. The $K_y$ ranges from 25811 $m^2 s^{-2}$ over land to 297053 $m^2 s^{-2}$ over sea. These diffusion coefficients tend to be slightly underestimated, because the variabilities of wind speed at higher frequencies (than 3-hourly) are not accounted for. This means that PHLET may underestimate the horizontal transport slightly.

**Appendix B. Satellite Conversion Matrix to account for the smoothing effect of satellite pixels**

The SCM is essentially a tool to preform quick conversion between grids, regular or not. In the YRD area, there are $100 \times 100 = 10000$ grid cells on the 0.05°×0.05° grid. We use the SCM (**A** matrix: [10000, 10000]) to convert from its original grid (**X** vector: [10000, 1]) to the final grid (**Y** vector: [10000, 1]), i.e., $\mathbf{Y} = \mathbf{A}\mathbf{X}$. The 10000 elements in one specific row of **A** represent the weights of the 10000 elements of **X** to an element in **Y**. Apparently, **A** is a sparse matrix. The following description shows how **A** is

constructed.

First, the VCDs specific to satellite pixels are reconstructed from the model grid cells. Each model grid cell (MGC) is divided into $10 \times 10 = 100$ finer grid cells (FGCs), each having the same area. Suppose the number of MGCs fully or partially covered by a given pixel p is $N_c$, and the number of FGCs in a given MGC i covered by p is $g_i^p$, then the total number of FGCs covered by p is:

$$G^p = \sum_{i=1}^{N_c} g_i^p \tag{B1}$$

Thus, the average VCD for the pixel p can be reconstructed as follows:

$$VCD^p = \sum_{i=1}^{N_c} \frac{g_i^p}{G^p} \cdot VCD_i \tag{B2}$$

Equation (B2) essentially means how a satellite pixel smooths the VCD. The blue portion of Fig. B denotes the projection from MGC i to pixel p.

10 The next step represents how the oversampling approach is applied to satellite-smoothed VCD data. Suppose the number of satellite pixels fully or partially covering an MGC j is Np, then the total number of FGCs being part of the intersection of the Np pixels and MGC j is:

$$G_j = \sum_{p=1}^{N_p} g_j^p \tag{B3}$$

Finally, the average VCD for the MGC j converted from the Np pixels is:

$$VCD'_j = \sum_{p=1}^{N_p} \frac{g_j^p}{G_j} \cdot VCD^p \tag{B4}$$

The pink portion of Fig. B denotes the projection from pixel p to MGC j. Thus, the element of SCM converting from MGC i to MGC j can be derived as follows:

$$A_{j,i} = \sum_{p=1}^{N_p} \frac{g_j^p}{G_j} \cdot \frac{g_i^p}{G^p} \tag{B5}$$

## Appendix C. Deriving NO₂ lifetime from VCD

We assume a steady state of radicals ($HO_x$), where the production rate of $HO_x$ is equal to the loss rate through three types of termination reactions: between the hydroxyl radical (OH) and $NO_2$, between NO and peroxyl radicals to form organic nitrates, and between peroxyl radicals (Murphy et al., 2006; Valin et al., 2011):

$$P(HO_x) = k_1 C_{OH} C_{NO_2} + \alpha k_{2eff} \frac{k_1 C_{NMVOC} C_{OH}}{k_{2eff} C_{NO}} C_{NO} + 6k_{3eff} (\frac{k_1 C_{NMVOC} C_{OH}}{k_{2eff} C_{NO}})^2 \tag{C1}$$

Here $P(HO_x)$ is the production rate, and the right-hand side of Eq. (C1) is the loss rate. $C_{NO_2}$ and $C_{OH}$ denote the concentrations of $NO_2$ and OH, respectively. Since the conversion between the peroxyl radicals ($HO_2 + RO_2$) and OH is in steady state, the term $\frac{k_1 C_{NMVOC} C_{OH}}{k_{2eff} C_{NO}}$ expresses the "effective" total concentration of peroxyl radicals in terms of the concentrations of NMVOC, OH and NO. Assuming $P(HO_x)$, $C_{NMVOC}$ and all reaction constants to be constant (Valin et al., 2011), and given that $C_{NO} = C_{NO_2} \cdot \frac{1-r}{r}$, Eq. (C1) can be simplified as Eq. (C2):

$$a C_{OH} C_{NO_2} + b C_{OH} + c(\frac{C_{OH}}{C_{NO_2}})^2 = 1 \tag{C2}$$

Here a, b, c are the coefficients. Because the chemical lifetimes of NO₂ is determined by $C_{OH}$ ($\tau_c = \frac{1}{k_1 C_{OH}}$), we can deduce the relationship between $C_{NO_2}$ and $\tau_c$:

$$\frac{a\prime}{\tau_c} + \frac{b\prime}{\tau_c \cdot C_{NO_2}} + c\prime(\frac{1}{\tau_c \cdot C_{NO_2}})^2 = -1 \tag{C3}$$

NO₂ is lost primarily through reaction with OH and secondarily through dry deposit ($\frac{C_{NO_2}}{\tau_d}$), thus its lifetime ($\tau$) is also determined by these two loss processes. Therefore,

$$\tau_c = \frac{1}{\frac{1}{\tau} - \frac{1}{\tau_d}} \tag{C4}$$

In the areas of low emissions, the emission term can be neglected in Eq. 2, thus the local net source $L = -\frac{C_{NO_2}}{r \cdot \tau}$. Therefore Eq. C3 becomes Eq. C5, which connects $L$ and $C_{NO_2}$.

$$a' \frac{r(L+kC_{NO_2})}{C_{NO_2}} + b' \frac{r(L+kC_{NO_2})}{C_{NO_2}^2} + c' (\frac{r(L+kC_{NO_2})}{C_{NO_2}^3})^2 = 1 \tag{C5}$$

where $k = \frac{1}{\tau_d}$. We determine the coefficients $a', b', c'$, and $k$ in Eq. C5 by conducing nonlinear fitting of

OMI NO$_2$ VCD data and $L$ values in the low emission areas (see below). This procedure establishes the nonlinear relationship between $\tau$ and VCD, which is then applied to the entire study domain.

The low-emission areas have small values of VCD and large negative values of $L$. Figure C shows a scatter plot for the derived local net source $L$ and VCD at each individual grid cell of the study domain. The data scatter reflects the combined effect of emission, loss, and horizontal transport. We then fit the

quantiles of $L$ where the VCD is relatively low ($< 5 \times 10^{15}$ molecules cm$^{-2}$, shown as blue points in Fig. C) into Eq. C5 through a nonlinear quantile fitter based on Tensorflow(Abadi et al., 2016). Using the quantile fitting also means that the low-emission grid cells do not need to be explicitly identified prior to the fitting. The quantile fitting gives $L$ as a function of VCD (when emissions are neglected), through which the relationship between lifetimes and VCDs is derived. We conduct the fitting by 50 times, each

by linearly changing the assumed percentile threshold of $L$ from 0.1% to 5%, to determine the fitted median value (red line in Fig. C) and uncertainty (gray shaded areas, 95% CI). The uncertainty is caused by the assumption on the NO$_2$/NO$_x$ ratio r, the simplification of the relationship between lifetimes and VCDs, and possible misjudgment of low-emission areas.

The orange line in Fig. C presents the relationship between NO$_2$ VCDs and chemical lifetimes ($\tau_c$) derived

based on the mean value of the fitting. The value of $\tau_c$ varies from 0.6 to 3.8 h with an average of 1.2 h. The lifetimes decline rapidly with increasing VCDs from 0 to $2 \times 10^{15}$ molecules cm$^{-2}$, and then grows gradually with increasing VCDs. This result is consistent with Valin et al. (2011). By comparison, the

value of $\tau_d$ is 30.4 h and is spatially homogeneous under the assumption here. The total lifetime ($\tau$) varies from 0.6 to 3.3 h (Fig. C, blue line) across the study domain.

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

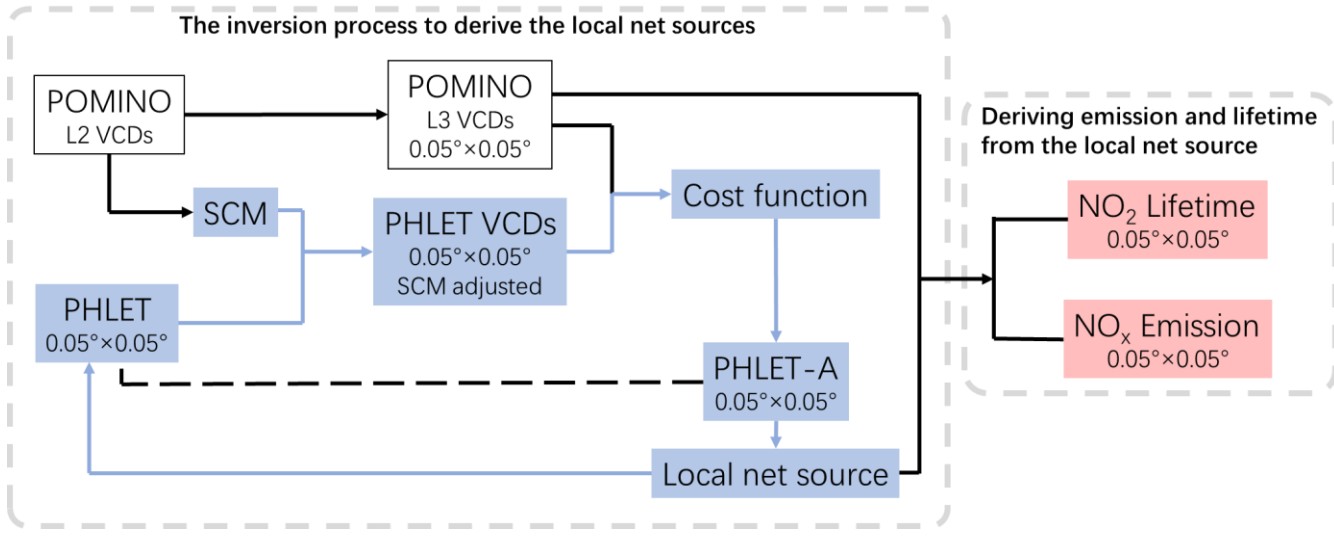

Figure 1. The flowchart of the framework of our methodology. The blue arrows show the iterative process.

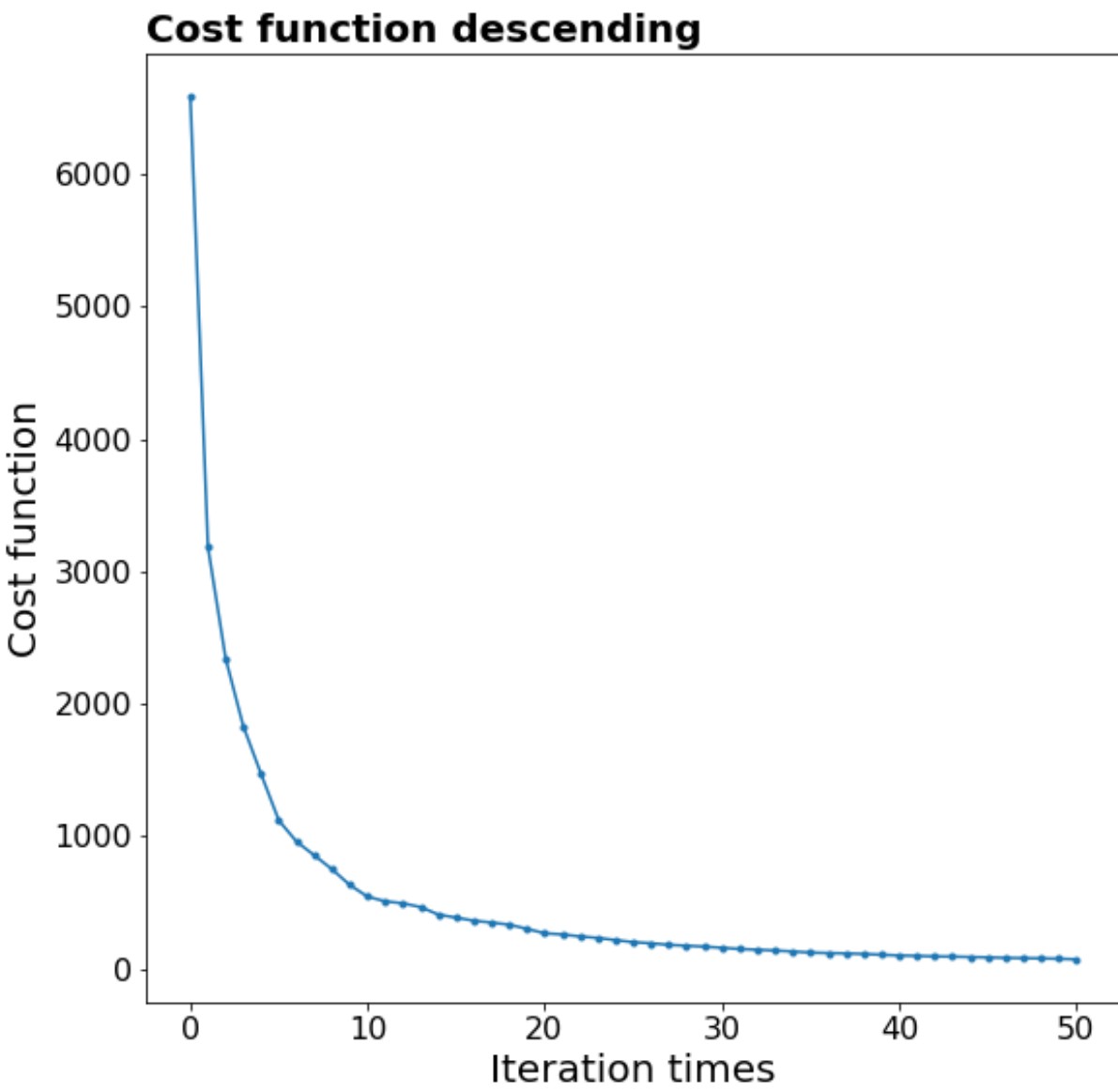

Figure 2. Cost function descending with iteration.

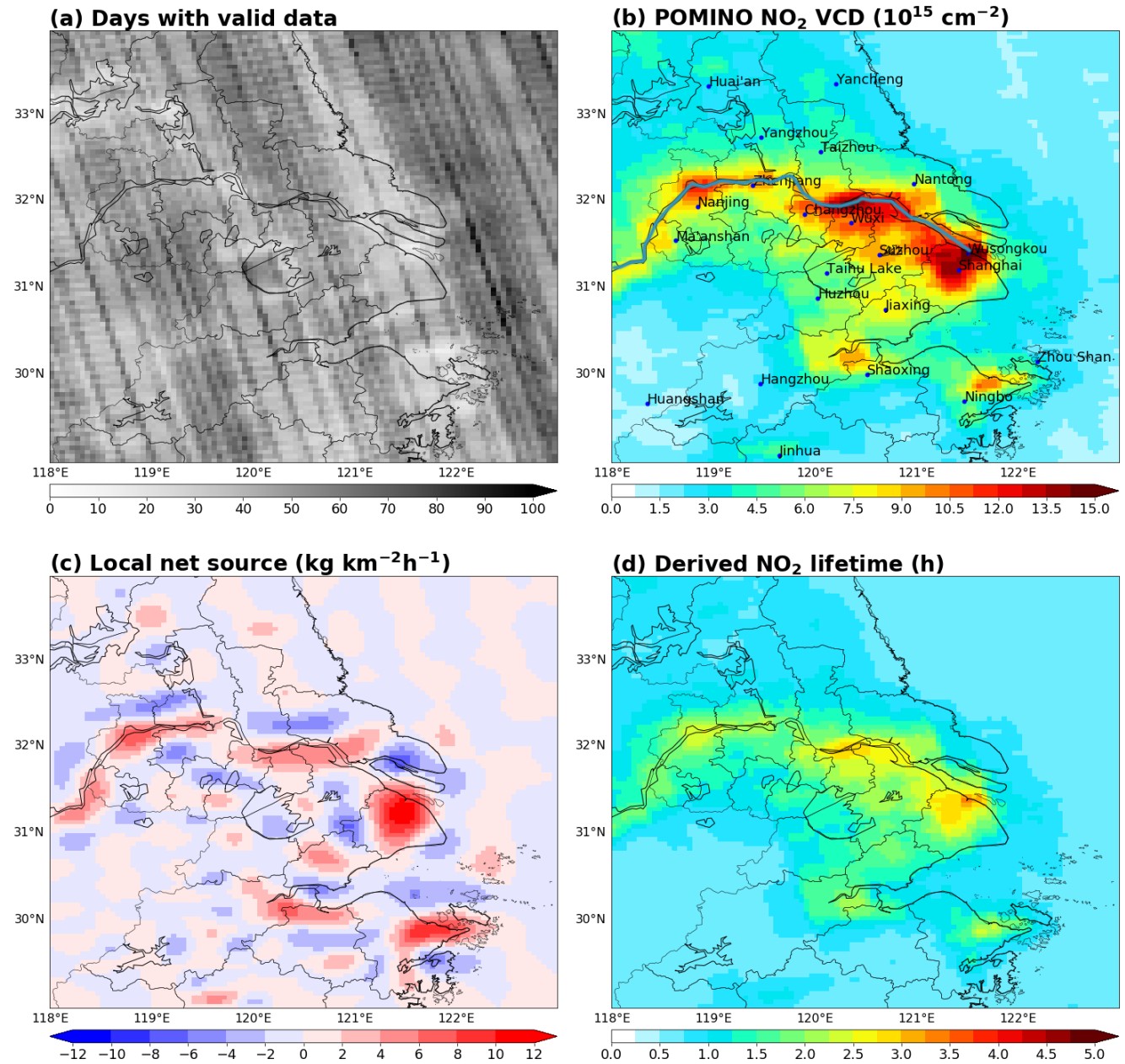

Figure 3. (a) Number of days with valid data in each grid cell over summer months of 2012–2015. (b) POMINO NO2 VCDs averaged over summer 2012–2015. Cities and locations mentioned in this paper are denoted. The Yangtze River is marked as a blue line. (c) Derived local net source. (d) Derived $NO_x$ lifetime due to both chemical loss and deposition.

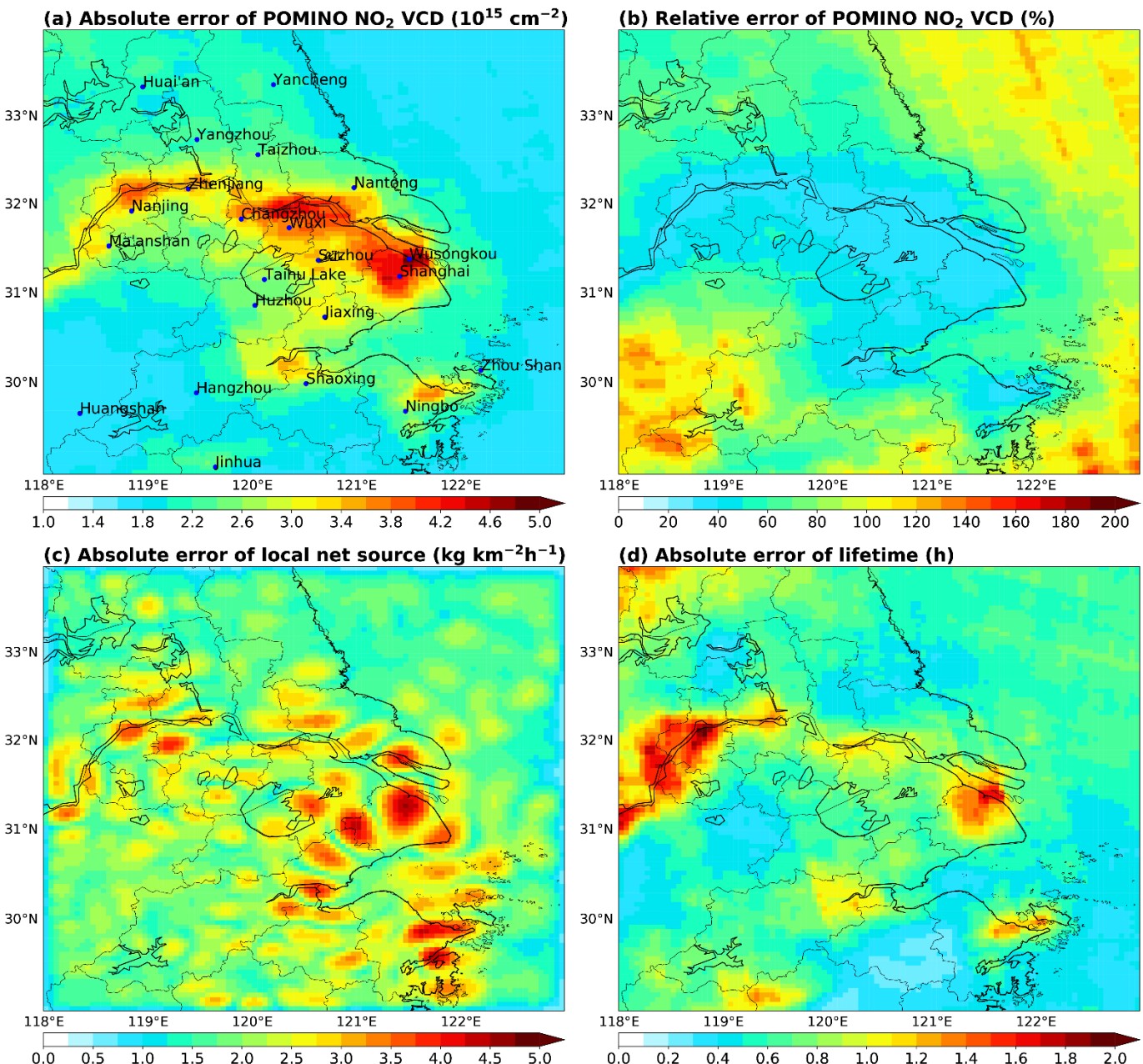

Figure 4. (a) Absolute error (1 σ) of POMINO NO₂ VCD at each grid cell at the 0.05°×0.05° resolution. (b) Relative error (1 σ) of POMINO NO₂ VCD. (c) Absolute error (1 σ) of the derived local net source. (d) Absolute error (1 σ) of the derived NOₓ lifetime due to chemical loss and deposition.

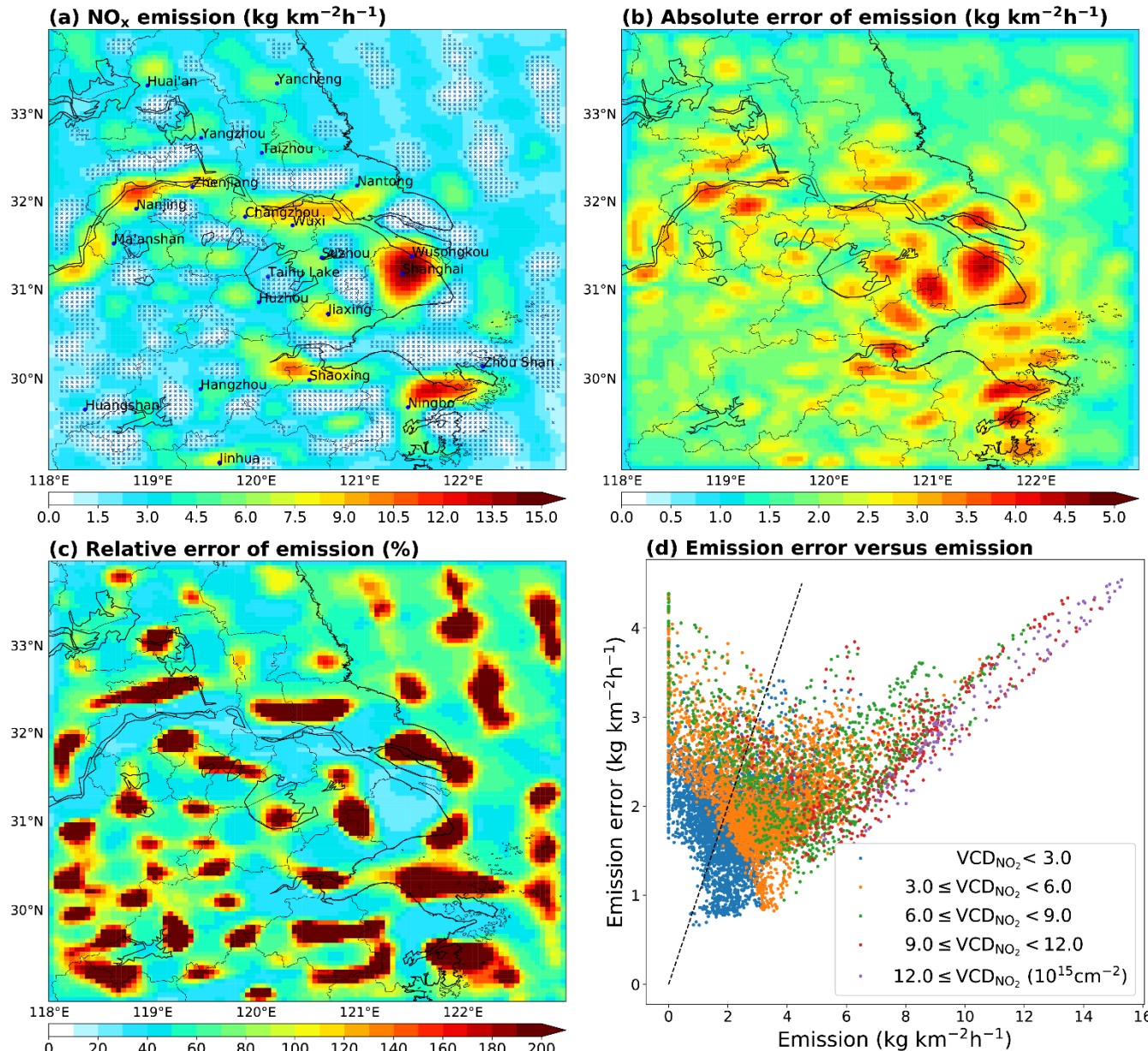

Figure 5. (a) Our NO$_x$ emissions from anthropogenic, biomass burning and soil sources together. The blue corsses indicate where the relative errors exceed 100%. (b) Absolute errors (1 σ) of our NO$_x$ emissions. (c) Relative errors of our emissions. (d) Absolute errors (1 σ) of NO$_x$ emissions as a function of NO$_x$ emissions at individual grid cells. Data points are coloured according to the magnitudes of POMINO NO$_2$ VCDs. The dashed line indicates an error of 100%.

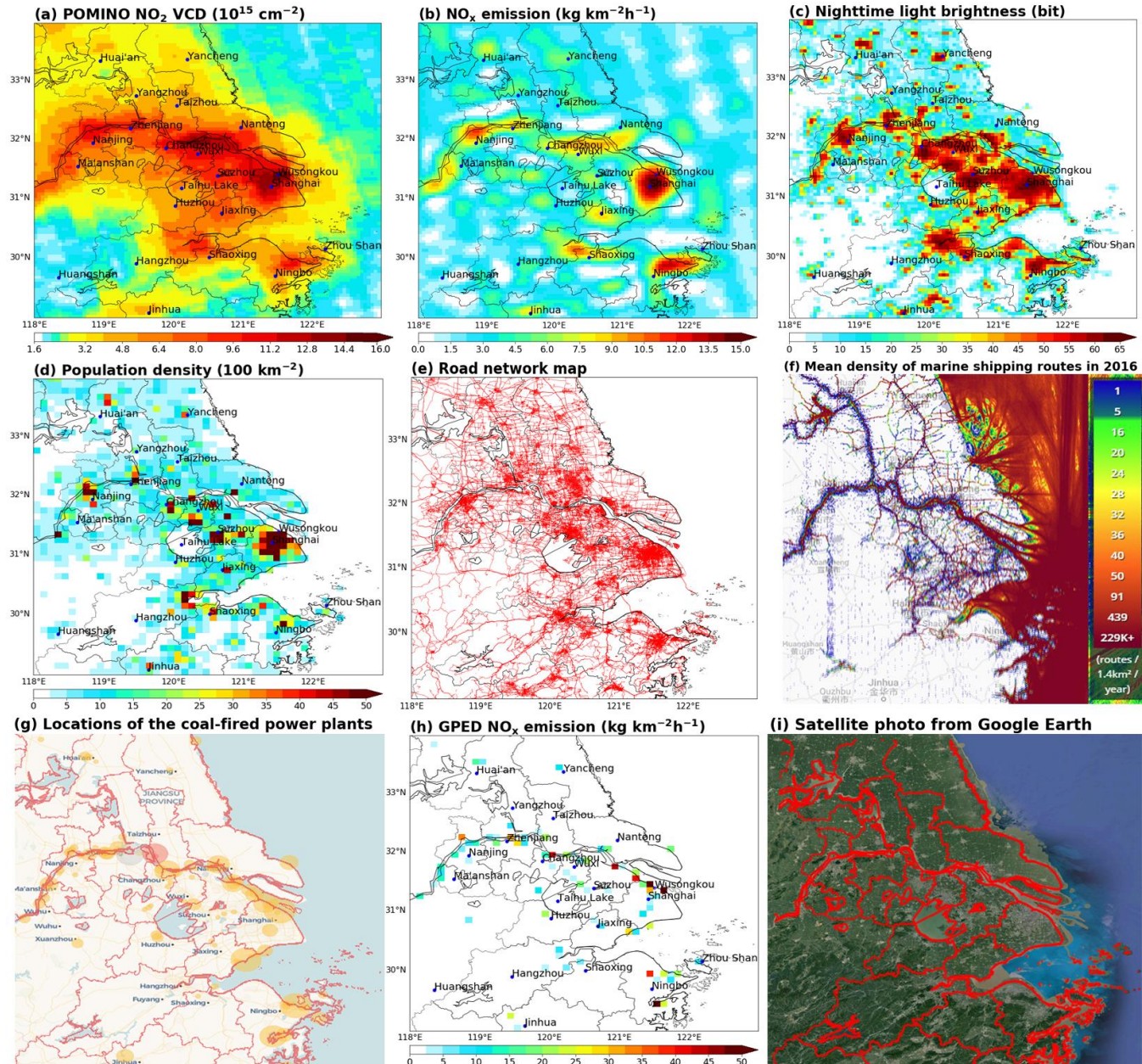

Figure 6. (a) POMINO NO₂ VCDs averaged over summer 2012–2015, which is same as Fig. 1a. but based on a different color scale. (b) Our NOₓ emissions from anthropogenic, biomass-burning and soil sources together, same as Fig. 3a. (c) Nighttime light brightness in 2012. (d) Population density averaged over 2012–2015. (e) Road network (red lines). (f) Mean density of marine shipping routes in 2016 (data source: www.marinetraffic.com). (g) Locations of coal-fired power plants in 2016 from Carbon Brief. (h) GPED

v1.0 bottom-up NO$_x$ emissions from coal-fired power plants. (i) A satellite photo from Google Earth taken in 2018.

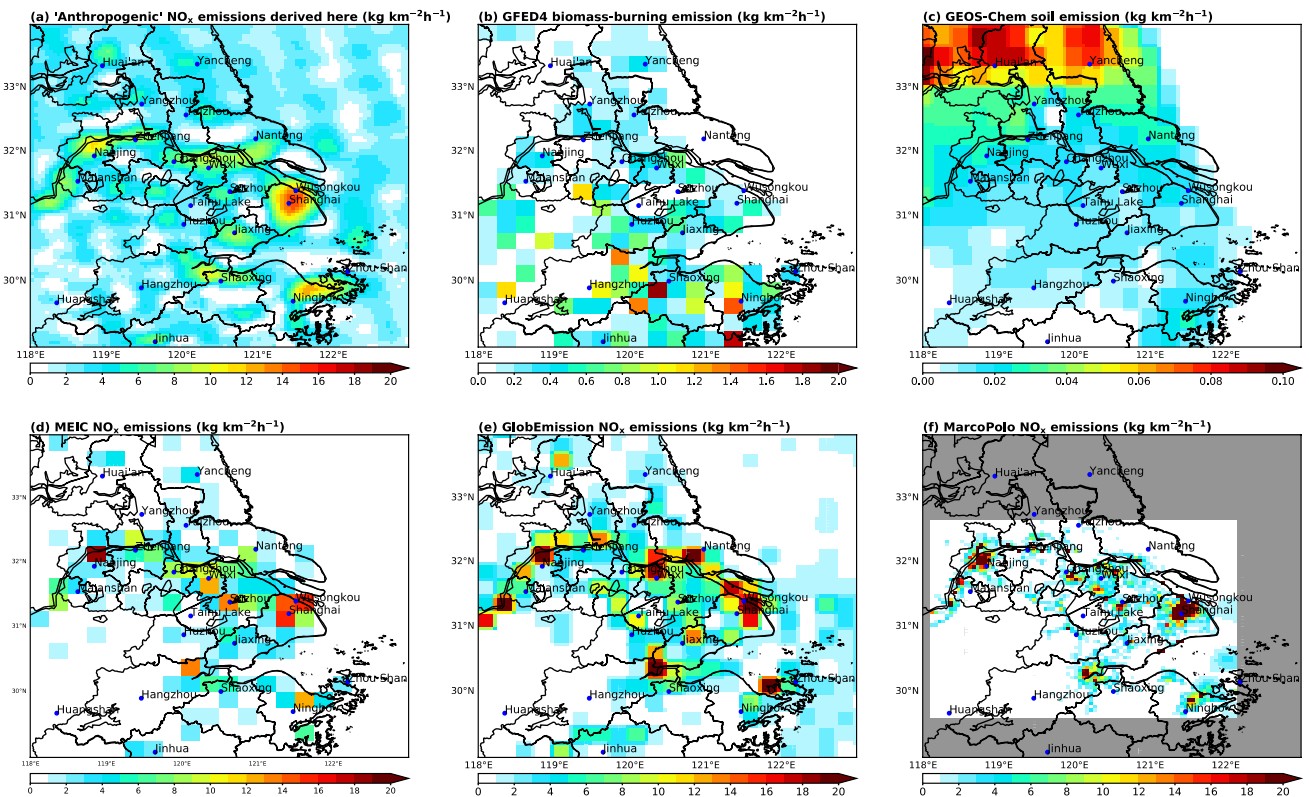

Figure 7. (a) Our "anthropogenic" NO$_x$ emissions, by subtracting soil and biomass burning emissions from the derived emissions. (b) GFED4 biomass burning NO$_x$ emissions. (c) Soil NO$_x$ emissions calculated by a nested GEOS-Chem simulation. (d) MEIC NO$_x$ emissions over summer 2012–2015. (e) DECSO v5.1qa top-down emissions in summer 2012–2015. (f) MarcoPolo bottom-up inventory in summer 2014; note that this inventory does not cover the grid cells shown in grey). All data are regridded to $0.05° \times 0.05°$.

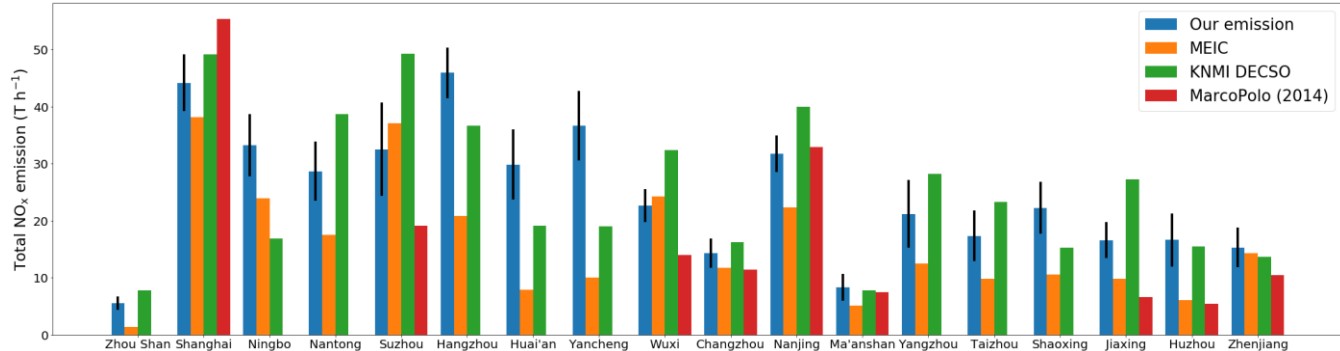

Figure 8. Total anthropogenic NO$_x$ emission in each city for summer 2012–2015 derived here, in comparison with other emission datasets. Soil NO$_x$ emissions calculated by the nested GEOS-Chem and biomass burning NO$_x$ emissions from GFED4 have been subtracted from our emission data and DECSO. Black vertical lines denote the uncertainty (1 σ) of our emissions.

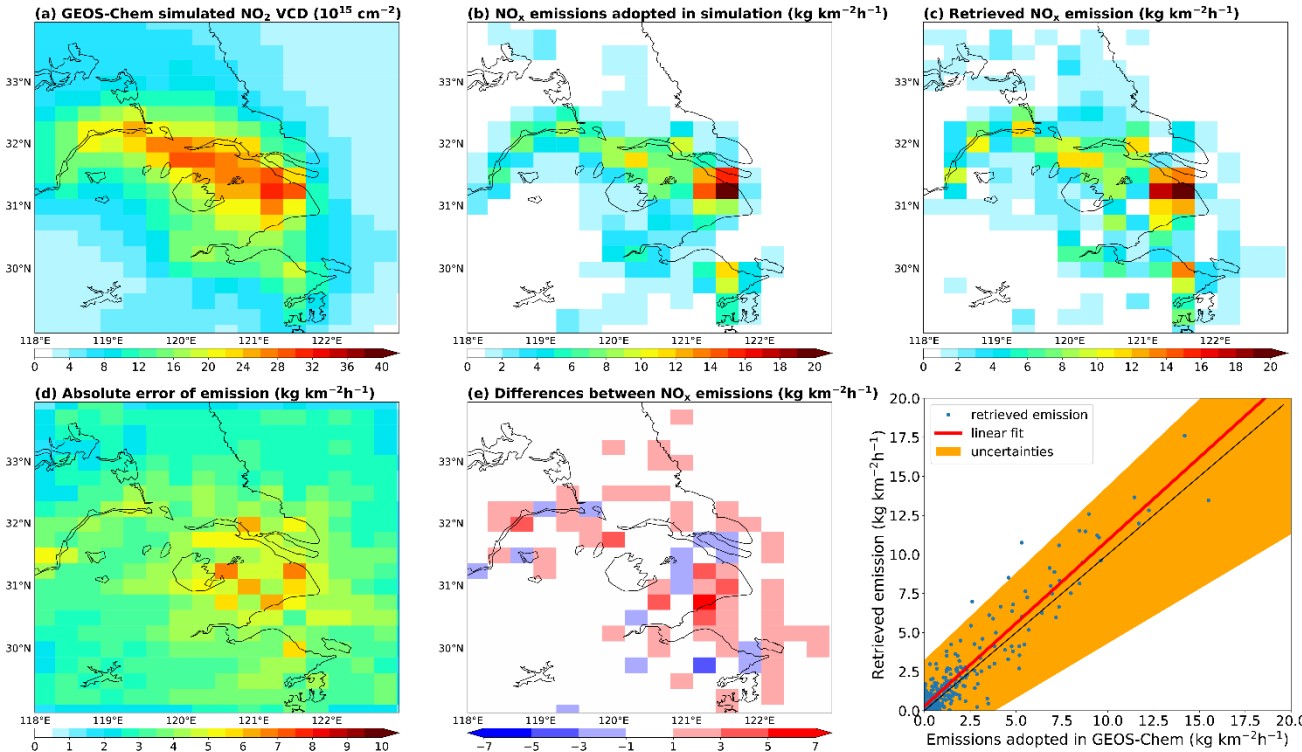

Figure 9. (a) GEOS-Chem simulated $NO_2$ VCDs at the $0.3125° \times 0.25°$ resolution for summer 2014. (b) Anthropogenic $NO_x$ emissions used in GEOS-Chem. (c) Anthropogenic emissions derived based on GEOS-Chem simulated $NO_2$ VCDs and our inversion method. (d) Absolute errors (1 σ) of derived emission data. (e) Differences between the derived emissions and GEOS-Chem emissions (derived minus GEOS-Chem). (f) Scatter plot for the derived emissions (y-axis) and GEOS-Chem emissions (x-axis). The red line represents least square linear fitting. The shading represents the fitting by accounting for errors in the derived emission data, i.e., derived emissions + 1 σ for the upper bound, and derived emissions – 1 σ for the lower bound. The black dotted line denotes the 1:1 line.

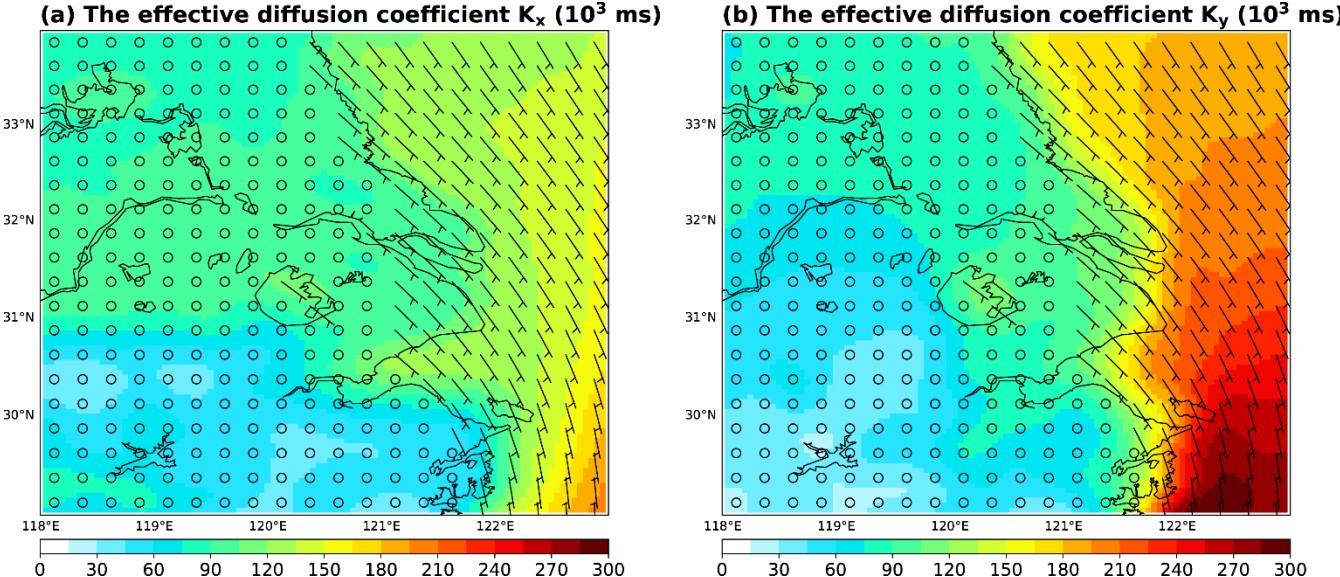

Figure A. The effective diffusion coefficients for summer 2012–2015 on a 0.05° × 0.05° grid. Overlaid is the temporal mean wind vector, which is plotted for every 5 × 5 = 25 grid cells to enhance the readability.

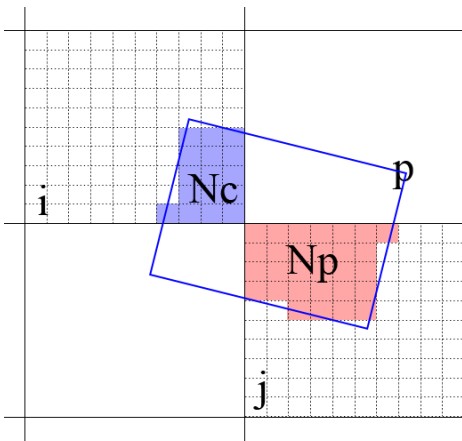

Figure B. Illustration of how the original PHLET model grid cell i is projected to the satellite pixel p and then to the final grid cell j through the SCM approach. The size of the satellite pixel is scaled down to be comparable with the size of a model grid cell, for illustration purposes.

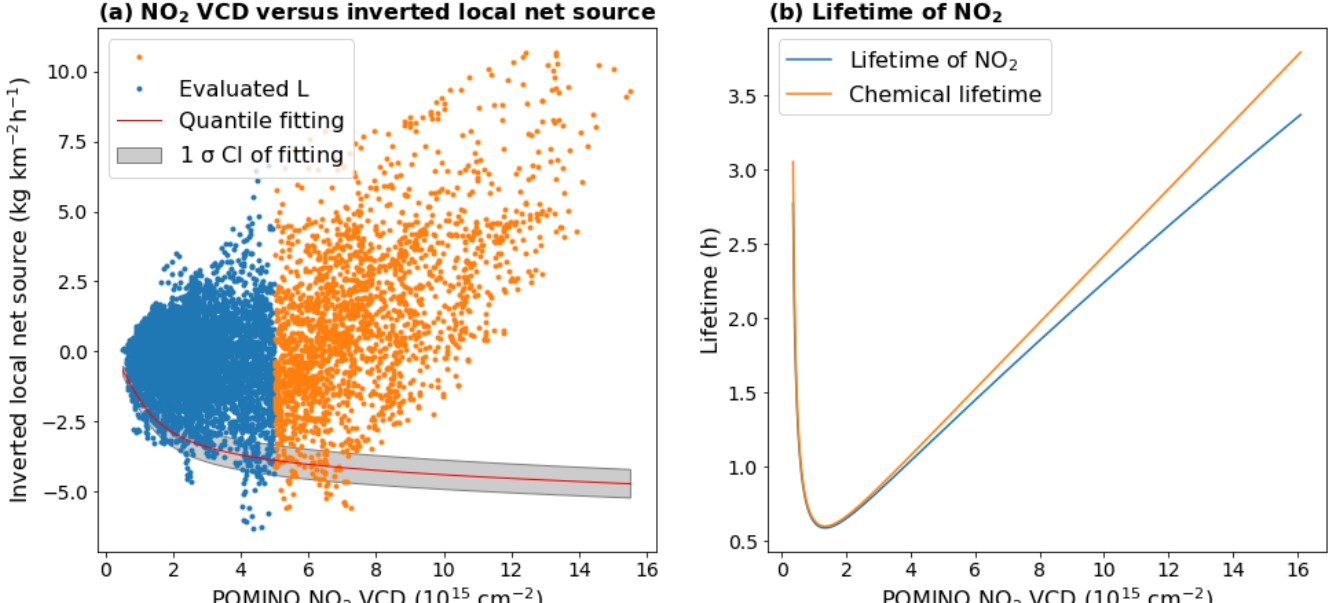

Figure C. (a) Scatter plot for POMINO $NO_2$ VCDs (x-axis) and derived local net sources (y-axis) across individual grid cells. Grid cells with $NO_2$ VCDs below (above) $5 \times 10^{15}$ molecules cm$^{-2}$ are coloured in blue (orange). The red line and shading denote the median and uncertainty (1 $\sigma$ CI) of the quantile fitting, respectively, to estimate the nonlinear relationship between $NO_2$ VCD and lifetime, based on data in the low-emission areas. (b) The derived relationship between $NO_2$ VCD and lifetime across the range of $NO_2$ VCDs in the YRD area.