# Peer review of "High-resolution ( $0.05^\circ \times 0.05^\circ$ ) $\text{NO}_x$ emissions in the Yangtze River Delta inferred from OMI"

_Atmospheric Chemistry and Physics, 2018_

## Referee Comment (RC1) · Anonymous Referee #1 · 17 Feb 2019

General comments: The manuscript entitled 'High-resolution (0.05°×0.05°) NOx emissions in the Yangtze River Delta inferred from OMI' focuses on developing a method to inverting NOx emissions at a high resolution in major urban areas by using the long-term satellite measurements of nitrogen dioxide. The results show that the inverted NOx emission dataset can reveal the features which are not well represented or not included widely used Multi-scale Emissions Inventory of China. Overall, though the topic is important and the methods are technically, the manuscript need be restructured and rephrased. I recommend to reconsider its publication pending the following concerns satisfactorily addressed.

Specific comments: 1. Why the shortest lifetime of NO2 has the advantage to better relate NOx emissions to NO2 VCDs at the 0.05°×0.05° resolution? 2. Page 6, Line

1-7: What's the relation between the NO2 retrieval with the AOD? The description is needed. 3. Section 2 is generally messy and lack of logics. What's the relation between the PHLET model and PHLET-A model? I suggest the authors rephrase the part 'data and method' more logically. 4. The main of this manuscript includes two parts: part one is to show the distributions of NO2 basing on the retrieved emission data, part two is to evaluate above emission data. Thus, showing more explicit analyses are needed. 5. In Figure 1, why the NOx emission and local net source are somewhat related to the lifetime of NO2? The good relationship between the NO2 VCD and lifetime of NO2 can be understood well, however, the relations with NOx emission and local net source are not taken for granted. 6. Figure 1 and Figure 2 should be rearranged. Fig. 2a-d can be combined into Fig.1a-d; Fig. 2e-f and Fig. 1f can be combined into one graph. The current arrangement is messy to describe. 7. Page 17, Line 6, what does 'Figure 3ows' mean? 8. How do the authors define 'anthropogenic' emission? Including what? 9. What's the reason of inconsistent difference of total anthropogenic NOx emission in each city for summer inverted by this study versus from the MEIC inventory? Otherwise, the difference should be same for each city, that is to say, systematically higher or lower. 10. The tile of Section 4.3 should be 'Comparing our inverted emission dataset with the MEIC inventory', or more exactly, it should be 'Comparison between our inverted emission dataset with the MEIC inventory'.

---

## Referee Comment (RC2) · Anonymous Referee #2 · 18 Apr 2019

**Review "High-resolution (0.05°x0.05°) NOx emissions in the Yangtze River delta inferred from OMI"**

The manuscript entitled 'High-resolution (0.05◦×0.05◦) NOx emissions in the Yangtze River Delta inferred from OMI' presents an inventive method to estimate NOx emissions on a high resolution from satellite observations. The topic is interesting and important. Important aspect is also the error analysis that has been described in detail. However, before the paper gets published, the manuscript can be improved a lot by adding more discussion of the results including the relation to other existing inventories, and the description of method should be rephrased in a clearer way.

**General remarks**

This inventory is presented as the only high resolution inventory for the YRD region, but in the MarcoPolo-Panda project, a high resolution emission inventory of 0.01 degree resolution has been developed for this region (see http://www.marcopolo-panda.eu/products/toolbox/emission-data/) Since the affiliations of the authors were also participating in the MarcoPolo-Panda project it is surprising that this inventory is not mentioned or used in their comparisons.

Also Zhao et al. (2015) present a city-scale emission inventory with the resolution of 3kmx3km in Nanjing, in the Yangtze River Delta.:

*Zhao, Y., Qiu, L. P., Xu, R. Y., Xie, F. J., Zhang, Q., Yu, Y. Y., Nielsen, C. P., Qin, H. X., Wang, H. K., Wu, X. C., Li, W. Q., and Zhang, J.: Advantages of a city-scale emission inventory for urban air quality research and policy: the case of Nanjing, a typical industrial city in the Yangtze River Delta, China, Atmos. Chem. Phys., 15, 12623-12644, 10.5194/acp-15-12623-2015, 2015*

The authors claim that very high resolution emissions are lacking, but it is not mentioned how "very high" is defined. Several regions have a high resolution emission inventory with similar resolution as PHLET: CAMS in Europe, GlobEmissions in the Qatar and South Africa, MarcoPolo-Panda in various regions in China.

On page 3, Line 13-20 other methods are shortly introduced in a very confusing way. The conclusions the authors make here are not always correct. I will be more specific:

- The authors mention that their method is not computationally expensive and can be applied world-wide, but a rough calculation shows that their algorithm will take at least 10 year to calculate the emissions for the whole world, which is not faster than many other methods they refer to op page 3.
- The authors say that the methods are limited in time period, spatial domain and horizontal resolution. This is very different for all the referred methods. The methods of Miyazaki et al and of Stavrakou et al. have already been applied on a global scale, while other methods are also not theoretically limited to a certain domain. In general, the methods mentioned can be applied to any time period as long as satellite observations are available.
- The authors suggest that only (Lin et al., 2012) and Stavrakou et al. (2013) provide uncertainties of the CTM, while they are also presented by Miyazaki et al. (2012) and Ding et al. (2017).

The paper is missing a discussion on the results, there is a section "concluding remarks" which gives a summary, but I miss the following discussions:

- What are pros and cons of the introduced method? The pros are mentioned, but what is the downside of averaging a time period of 2012-2016. Specifically in this period strong trends are appearing in NO2 over China.
- The period is focusing on the summer time. What are the expected results for the winter-period. Will this change the spatial resolution? Will the magnitude of the emissions change a lot?
- It should at least be mentioned that there is no sector information in the derived emissions, which is an advantage of bottom-up inventories
- Although the gridding is on a 0.05° resolution the actual spatial resolution of the resulting emissions seems much lower. An indication of the intrinsic resolution can be obtained from the largest gradients in the emissions. I cannot detect clear structures with a 0.05° resolution. One would at least expect some power plants to show up as clear spots in this region. Maybe the method is still limited by the OMI resolution?

The structure of the paper is somewhat confusing and therefore I suggest moving appendix C and E to the main text. Section 2.1 is too short to understand the method of determining emissions.

In section 2.3: The contributions of lightning, biomass burning and aircraft emissions are neglected. The authors explained that the contributions of these emissions are small. In the inversion method, both soil and anthropogenic emissions are derived. In section 4, it is calculated that the soil emissions contribute 0.9% of the inverted emissions. This looks like a very small amount. However, biomass burning is considered as a significant source in the YRD, especially in summer. On a scale of $0.05° \times 0.05°$ lots of biomass burning activity will exist. Give some more detailed information and explain why lightning, biomass burning and aircraft emissions are neglected.

The model error is set to be the sum of the quadrature of errors contributed by several aspects. However, there is no explanation on how the authors set some errors, for example the treatment of background NO2 concentrations. The authors use wind fields from ECMWF on a coarse resolution and regridded to a high resolution. The error of regridded wind field on high resolution can be quite large. The authors consider error of wind speed, but how about the wind direction? The error set for the wind looks optimistic.

References: All the references should be carefully checked if they are in the correct format, especially the names of authors.

Many articles are missing or articles should be removed throughout the whole text. It is advisable to let a native speaker make the necessary corrections.

**Detailed comments**

Page 1, Line 14: lacking => missing

Page 1, Line 17: The inversion => The model used in the inversion

Page 1, Line 18: We construct a model called PHLET (..)

Page 1, Line 19 Metrix => Matrix

Page 2, Line 5 tied => related

Page 2, Line 7 features => structures

Page 2, Line 8-9: This last sentence is kind of obvious. It should be moved from the abstract to the conclusions/outlook, but I suggest to just remove it.

Page 2, Line 21: split the sentence into 2 separate sentences to make it more understandable.

Page 2, line 24: on => of

Page 2, line 24: how is a "very high resolution" defined?

Page 3, Line 1: Bottom-up emissions do not only use spatial proxies but are also based on gathered statistical information of industrial output, car emissions, etc.

Page 3, Line 4: Please define "high"

Page 3, Line 20-21: What do the authors mean by "low-cost" and "high-resolution"

Page 3, Line 22: Here it is mentioned that these inventories are important for trends and variability. I agree, but the method presented in this paper do not give the possibility to study trends and variability, which should be mentioned somewhere in the conclusion.

Page 3, Line 22-23: Why is it important to understand air pollution with the advent of TROPOMI? I would say it is the other way around: TROPOMI is important for understanding air pollution.

Page 3, line 24: Constructing => construct

Page 4, line 1 other 13 => 13 other

Page 4, line 1: explain the acronym POMINO

Page 4, line 3: change to "a model called PHLET"

Page 4, line 5: delete "concentration dependent"

Page 4, line 17: Why is this the finest spatial information possible?

Page 4, line 19: What Is SCM? This is explained much later in the text.

Page 4, line 21-22: Without further explanation this does not explain the method.

Page 4, line 22: Which fixed formula is used?

Page 5, line 1: Which nonlinear relationship do the authors mean here? There are 3 quantities mentioned: (1) emissions , (2) lifetimes and (3) VCDs.

Page 5, line 4: A long time period is mentioned. What do the authors mean, a long time period to average or multiple 5 years periods? And why are these long time periods not presented in this paper?

Page 5, line 5: It is mentioned that the calculation takes about 36 hours after necessary input data? What are the necessary input data? How long does it take to prepare the input data?

Page 5, line 5: If the inversion takes 36 hours for a 5x5 degree domain, a global calculation will take about 10000 hours, which is about 10 year.

Page 5, line 8: a reference for OMI is missing.

Page 6, line 2-7: Removing the 30 outer pixel and the row anomaly will strongly reduce the number of pixels used in this research. How may pixels are still used?

Page 6, line 8: space => grid

Page 6, line 11: The footprint does not change, the location of the footprint changes from one day to another.

Page 6, line 14: the year of the reference to Fioletov is missing.

Page 6, line 17: For purpose => For the purpose

Page 7, Line 7: The assumptions of the PHLET model are not mentioned in Beirle et al. This reference should be removed.

Page 7, Line 10: The transport from neighbouring regions is missing in this list while this is an important contribution.

Page 7, Line15-16: Can the authors give a reference for this statement.

Page 7, Line 17-20: This is quite some assumption about the background value. What is the basis of this assumption? Why is the uncertainty set to 5%?

Page 8, line 14: What is the source of the wind data?

Page 9, Line 20: space => grid

Page 11, Line 7-8: I would suggest mentioning the average number of iterations (about 60?) needed to reach convergence and remove Fig C1. Is the value of 390 chosen based on this Figure and the fact that it is stable or are there other motivations?

Page 12, Line 6: It becomes more clear if the short appendix C is just put into the text here.

Page 12, Line 11: "inverted emission" is not the correct term. The concentrations are inverted to get the emissions. This "emission inversion" and "inverted emissions" is appearing in many places in the text.

Page 12, Line 11-12: What is the value of error on the lifetime? I suggest mentioning also the values of the calculated errors in the text.

Page 14, line 1: inverted => derived

Page 14, line 14: Since there is a lot of agriculture in the YRD region a soil contribution of 0.9 % seems very small and needs some explanation. A discussion on biomass burning emissions (which occur in the agricultural regions) can be helpful here as well.

Page 15, Line 1: Figure 2e is mentioned without discussing 2a-d.

Page 15, Line 13: Please mention the basis of the coloring of Fig 2f.

Page 15, Line 19: Why are the emissions not directly compared to bottom-up inventories instead of these proxies that are used in the bottom-up inventories. For example a comparison with the MarcoPolo-Panda or the Zhao et al. inventory can give more insights.

Page 17, Line 5-6: To separate the anthropogenic emissions, GEOS-Chem is used to calculate soil emissions. What is the uncertainties of the soil emissions calculated by GEOS-Chem?

Page 17, Line 6: Figure 3ows?

Page 17, Line 8: Comparing Figure 3e and 3f is only useful if they are at the same resolution. Thus Figure 3e should be regrided to the coarser resolution of Figure 3f.

Page 18, Lines 6-15: There is some repetition of the text of the previous sections

Page 18, Line 19-20: I would remove the last sentence about the programming language, which is not very relevant in a scientific paper.

Page 22: Line 17: Since one observation of the satellite is used in several grid cells, I doubt if the assumption that covariance matrices are diagonal matrices is correct. A discussion is needed here.

References: Most references contain many spelling errors and omissions.

Figure 2d: The lifetime is very short over the ocean, contradicting to what is usually seen in the literature.

Figure 2e: Although the gridding is on 0.05 degree resolution the actual spatial resolution of this image seems much lower. An indication if the intrinsic resolution can be obtained from the largest gradients in the emissions. I cannot detect clear structures with a 0.05 degree resolution. One would at least expect some power plants to show up as clear spots in this region.

Maybe the method is still limited to the OMI resolution. I would like to see some discussion about this.

Figure 2f: The plot is more logical when the x-axis and y-axis are reversed. I also suggest drawing a line for the 100% relative error in this plot as a helpline to guide the eye.

Figure 2 caption: What are the magnitudes of POMINO that are mentioned. In Figure 2f it is too small to see.

Figure 3e: This has to be regrided to the resolution of 2f for comparison.

Figure 3: I miss information on power plants, which are a major source of NOx.

Figure 4: The crosses in the plot, indicating the amount of grid cells, are too small.

Figure D1: I do not understand why we need two colors for the dots.

Figure D2: The lifetime depends a lot on chemistry, temperature and precipitation. Therefore, the plot seems very simplified.

---

## Author Comment (AC1) · 7 Aug 2019

Thanks for your insightful comments and suggestions. We have revised our manuscript accordingly. Please check the supplementary for the revised paper and the response.

Please also note the supplement to this comment:
https://www.atmos-chem-phys-discuss.net/acp-2018-1275/acp-2018-1275-AC1-supplement.zip

---

## Author Response (AR1)

General Comments

1. The manuscript entitled 'High-resolution (0.05°×0.05°) $NO_x$ emissions in the Yangtze River Delta inferred from OMI' focuses on developing a method to inverting $NO_x$ emissions at a high resolution in major urban areas by using the long-term satellite measurements of nitrogen dioxide. The results show that the inverted $NO_x$ emission dataset can reveal the features which are not well represented or not included widely used Multi-scale Emissions Inventory of China. Overall, though the topic is important and the methods are technically, the manuscript need be restructured and rephrased. I recommend to reconsider its publication pending the following concerns satisfactorily addressed.

The manuscript has been overhauled considering the comments from both referees.

A brief review has been made about the inventories at similar resolutions, including Zhao et al. (2015) and CAMS-reg (Granier et al., 2019) on page 3 line 8 (see the revised manuscript) based on bottom-up methods. Top-down estimates can be further combined with bottom-up inventories and spatial proxies to increase the spatial resolution, by downscaling and/or source sector apportionment (e.g., MarcoPolo on page 4 line 1-4). MarcoPolo emissions can reach higher resolutions than 0.05°×0.05°, i.e., 0.01°×0.01°, given the detailed information of the location of the emission sources which ask for lots of efforts to collect and are absent or inaccurate at times. Top-down emissions including our work offer an important supplement and reference at high resolutions.

The PHLET model has been upgraded and re-built on the FEniCS platform, the necessary citations of which have also been included. Based the FEniCS platform, we improve the calculation efficiency of the PHLET and A-PHLET largely. Now, the inversion calculation takes less than one hour as stated on page 6 line 14.

We have also fixed a bug to correctly account for the effect of $S_0$ in Eq. 3. The corresponding results and discussions have been revised including $NO_x$ emissions, lifetimes and the uncertainties. After processing the error covariance properly, the derived the lifetime of $NO_2$ due to deposition becomes longer (30.4 h), which is more consistent with our knowledge about $NO_x$ chemistry.

We have shortened the study time period from summer 2012-2016 to summer 2012-2015. According to the National Bureau of Statistics of China (http://data.stats.gov.cn/), $NO_x$ emissions have dropped substantially from 2015 to 2016. Thus, including summer 2016 may not be the best practice to derive emissions.

To substantiate the emission distribution, more discussion has added in Sect. 4.2 based

on the distributions of proxies such as nighttime light, population density, marine shipping routes, coal power plant locations and land use indicated by a satellite photo from Google Earth. Sect. 4.3 compares our emissions with other inventories besides MEIC.

In the conclusion section, we give a summary of the limitations and shortcomings of our method.

Most of the figures have been re-arranged. Some figures have been added, considering comments from both referees.

We have substantially improved the structure of the manuscript to accommodate both reviewers' suggestions. A flowchart has been added to Sect. 2.1 in order to illustrate the procedures of our inversion method. Section 2.3 has been divided into 5 subsections for clarification. Section 2.3.1-2.3.3 describe the model setting and assumptions. Sect. 2.3.4 shows how the SCM matrix is applied to PHLET simulated VCDs, with the detailed procedures shown in Appendix B. Section 2.3.5 summarizes the uncertainty estimates. The part (former Appendix D) about solving the observation error covariance matrix and the adjoint model has been moved to Sect. 2.4, supplemented with an extended discussion on assuming the covariance to be diagonal. The OSSE-like test (former Appendix E) based on GEOS-Chem simulated $NO_2$ data has been moved to a new Sect. 5.

Specific comments

1.  Why the shortest lifetime of $NO_2$ has the advantage to better relate $NO_x$ emissions to $NO_2$ VCDs at the 0.05°×0.05° resolution?

Due to the short lifetime of $NO_2$, the effect of transport and diffusion is rather local. Therefore, the distribution of $NO_2$ VCDs can better reflect that of $NO_x$ emission at high-resolution; and the effect of transport errors on emission estimate is smaller.

2.  Page 6, Line 1-7: What's the relation between the $NO_2$ retrieval with the AOD? The description is needed.

The $NO_2$ retrieval becomes unreliable when the loading of aerosol gets too high. We have added necessary citation to this description.

3.  Section 2 is generally messy and lack of logics. What's the relation between the PHLET model and PHLET-A model? I suggest the authors rephrase the part 'data and method' more logically.

In order to clarify our method, we have added a flowchart and additional descriptions to illustrate the procedures in Sect. 2. See our response to general comment 1 for the

detailed structural changes.

4. The main of this manuscript includes two parts: part one is to show the distributions of $NO_2$ basing on the retrieved emission data, part two is to evaluate above emission data. Thus, showing more explicit analyses are needed.

To substantiate the emission distribution, more discussion has added in Sect. 4.2 based on the distributions of proxies such as nighttime light, population density, marine shipping routes, coal power plant locations and land use indicated by a satellite photo from Google Earth. Sect. 4.3 compares our emissions with other inventories besides MEIC.

5. In Figure 1, why the $NO_x$ emission and local net source are somewhat related to the lifetime of $NO_2$? The good relationship between the $NO_2$ VCDs and lifetimes of $NO_2$ can be understood well, however, the relations with $NO_x$ emission and local net source are not taken for granted.

We have clarified the methodology; see our response to general comment 1.

As shown in Eq. (2), the local net source is the difference between emission and loss.

Sect. 2.5 and Appendix C presents how to calculate emission and lifetime from the local net source.

6. Figure 1 and Figure 2 should be rearranged. Fig. 2a-d can be combined into Fig.1a-d; Fig. 2e-f and Fig. 1f can be combined into one graph. The current arrangement is messy to describe.

More figures are included in the revised manuscript. The figures are also re-arranged taking the comments from both of the referees into consideration.

7. Page 17, Line 6, what does 'Figure 3ows' mean?

Typing error. Changed.

8. How do the authors define 'anthropogenic' emission? Including what?

As now clarified in Sect. 2.3.2 (page 10 line 13-21):

"Lightning emissions, biomass burning emissions, aircraft emissions, transport from neighboring regions, and convection can lead to $NO_2$ at higher altitudes over the YRD area. However, the amount of $NO_2$ aloft is much smaller than near-ground NO2 due to large ground sources (Lin, 2012). Thus, we regard $NO_2$ aloft as the regional background, and do not include it in Eq. 1. Also, for near-ground $NO_2$ over the YRD area, the

contribution of downward vertical transport is negligible compared to the contribution of ground sources. Aircraft emissions contribute little to the total ground source, because 78% of aircraft emissions occur at the high altitudes (9–12 km) (Ma and Xiuji, 2000). Therefore, PHLET only accounts for near-ground $NO_2$ from ground soil, biomass burning and anthropogenic sources (energy, industry, transportation, and residential).”

And in Sect. 4.3 (page 21 line 11-13):

“Our emission data and the DECSO inventory are top-down estimates and include the contributions of soil and biomass-burning sources. Thus, we estimate soil and biomass burning emissions from independent sources, and then subtract these emissions from our and DECSO emission datasets” (to obtain anthropogenic emissions.)

9. What's the reason of inconsistent difference of total anthropogenic NOx emission in each city for summer inverted by this study versus from the MEIC inventory? Otherwise, the difference should be same for each city, that is to say, systematically higher or lower.

Both our and MEIC inventories are gridded, and their differences are grid cell independent and vary from one city to another.

10. The tile of Section 4.3 should be 'Comparing our inverted emission dataset with the MEIC inventory', or more exactly, it should be 'Comparison between our inverted emission dataset with the MEIC inventory'.

Changed

**Anonymous Referee #2**

General Comments

1. This inventory is presented as the only high-resolution inventory for the YRD region, but in the MarcoPolo-Panda project, a high-resolution emission inventory of 0.01-degree resolution has been developed for this region (see http://www.marcopolo-panda.eu/products/toolbox/emission-data/) Since the affiliations of the authors were also participating in the MarcoPolo-Panda project it is surprising that this inventory is not mentioned or used in their comparisons. Also, Zhao et al. (2015) present a city-scale emission inventory with the resolution of 3kmx3km in Nanjing, in the Yangtze River Delta.: “*Zhao, Y., Qiu, L. P., Xu, R. Y., Xie, F. J., Zhang, Q., Yu, Y. Y., Nielsen, C.*

*P., Qin, H. X., Wang, H. K., Wu, X. C., Li, W. Q., and Zhang, J.: Advantages of a city-scale emission inventory for urban air quality research and policy: the case of Nanjing, a typical industrial city in the Yangtze River Delta, China, Atmos. Chem. Phys., 15, 12623-12644, 10.5194/acp-15-12623-2015, 2015.*" The authors claim that very high resolution emissions are lacking, but it is not mentioned how "very high" is defined. Several regions have a high-resolution emission inventory with similar resolution as PHLET: CAMS in Europe, GlobEmissions in the Qatar and South Africa, MarcoPolo-Panda in various regions in China.

We have included discussion of these inventories in the revised introduction:

On page 3 line 1-3: "Gridded bottom-up emission inventories typically use spatial proxies (like population and GDP) to allocate provincial-level emission values, which are derived from activity statistics and emission factor data, to individual locations (Zhao et al., 2011; Janssens-Maenhout et al., 2015; Zhao et al., 2015).".

On page 3 line 7-10: "For a small area, emission factors and activity data of the major sources can be collected by on-site surveys to allow construction of a high-resolution inventory (Zhao et al., 2015; Granier et al., 2019), such as Zhao et al. (2015) for Nanjing. However, on-site surveys are extremely time consuming and resource demanding, difficult to be applied to a large domain in a timely manner."

On page 4 line 1-10: "Top-down estimates can be further combined with bottom-up inventories and spatial proxies to increase the spatial resolution, such as from $0.25°\times0.25°$ in the DECSO derived emissions to $0.01°\times0.01°$ for 2014 during the MarcoPolo Project (Hooyberghs et al., 2016; Timmermans et al., 2016) and similar inventories over Qatar and South Africa (Maiheu and Veldeman, 2013)."

In this work, high resolution refers to emissions at a resolution equal or higher than $0.05°\times0.05°$. We have made this clear on page 2 line 23 and page 4 line 9.

2. The authors mention that their method is not computationally expensive and can be applied world-wide, but a rough calculation shows that their algorithm will take at least 10 year to calculate the emissions for the whole world, which is not faster than many other methods they refer to on page 3.

First, our method is designed for urban and surrounding areas, rather than everywhere of the globe. Second, in this case study, the calculation is completed on only one CPU core, while the CTMs adopted in top-down method generally ask for parallel computing with many cores. For a multi-domain study, our method can easily adopt parallel computation with more cores. Third, we have upgraded the codes on the FEniCS platform, the necessary citations of which are included. Right now, the emission calculation for the YRD takes less than one hour, faster than our previous calculation by a factor of 30-40. Thus, with one computational core, applying our method to the

globe on a 0.05°×0.05° grid for 4 years would take about a few months.

3.    The authors say that the methods are limited in time period, spatial domain and horizontal resolution. This is very different for all the referred methods. The methods of Miyazaki et al and of Stavrakou et al. have already been applied on a global scale, while other methods are also not theoretically limited to a certain domain. In general, the methods mentioned can be applied to any time period as long as satellite observations are available.

We have clarified this point in the revised introduction (page 3 line 25 – page 4 line 1):

"These more sophisticated methods have often been applied to relatively short time periods (e.g., Gu et al., 2016 for one month), small spatial domains (e.g., Tang et al., 2013 in Texas), and/or at coarse horizontal resolutions (e.g., Miyazaki et al., 2012 at 2.8° and Stavrakou et al., 2008 at 5°×5°)."

4.    The authors suggest that only (Lin et al., 2012) and Stavrakou et al. (2013) provide uncertainties of the CTM, while they are also presented by Miyazaki et al. (2012) and Ding et al. (2017).

The statement and the citations are modified on page 4 line 6-8: "CTM-based studies typically provide an estimate of the overall model error, although Lin et al. (2012) and Stavrakou et al. (2013) present errors in the individual model processes (e.g., key chemical reactions and meteorological parameters)."

5.    What are pros and cons of the introduced method? The pros are mentioned, but what is the downside of averaging a time period of 2012-2016. Specifically, in this period strong trends are appearing in $NO_2$ over China.

We have shortened the study time period from summer 2012-2016 to summer 2012-2015. According to the National Bureau of Statistics of China (http://data.stats.gov.cn/), $NO_x$ emissions have dropped substantially from 2015 to 2016. Thus, including summer 2016 may not be the best practice to derive emissions.

We have also evaluated the emission for each year from 2012 to 2015. The average emissions of those years over 2012-2015 and the emissions evaluated from all VCD data together (as in the main text) are similar. Slope and interception of their linear regression are 0.95 and 0.08, and their correlation coefficient is 0.98 (see the figure below).

We discussion the limitation of our method in Section 2.

On page 9 line 9-11: "Also, combining data from multiple years to derive an averaged $NO_2$ distribution for simulation (rather than conducting the simulations for individual

years and months) leads to an additional uncertainty."

We have also added the revised conclusion section a paragraph summarizing the limitations of our method:

On page 25 line 11-20: "Our inversion method also has a few shortcomings. The derived emissions do not separate the individual contributions of anthropogenic sectors (i.e., power plants, industry, transportation, and residential). The spatial resolution of the estimated emissions is limited by that of satellite VCD data, although a special oversampling technique has been used to help achieve the highest spatial resolution possible for emissions. The PHLET model is assumed to be 2-dimensional by simplifying the vertical distribution of $NO_2$ and not accounting for the spatial variability in the vertical shape, similar to previous studies. The adjoint model assumes the observational error covariance matrix to be diagonal, without fully considering the effect of correlations between individual grid cells. Also, we assume a spatially uniform relationship between $NO_2$ VCDs and $NO_2$ lifetimes, which may lead to an underestimate in the lifetimes at low-$NO_2$ locations over the eastern sea."

[Figure]

6. The period is focusing on the summer time. What are the expected results for the

winter-period. Will this change the spatial resolution? Will the magnitude of the emissions change a lot?

To achieve highest spatial resolution possible, we have intended to focus on the summer months, when the lifetimes of $NO_x$ are the shortest.

The lifetimes of $NO_2$ would be longer in winter, and therefore, the effects of transport and diffusion are more significant. The spatial relation of $NO_2$ VCDs and $NO_x$ emissions would be lower in winter. Thus, it would be much more difficult to derive the emissions at a high resolution, and the influences of transport errors would be much larger.

7.  It should at least be mentioned that there is no sector information in the derived emissions, which is an advantage of bottom-up inventories

We have added on page 25 line 11-12: "Our inversion method also has a few shortcomings. The derived emissions do not separate the individual contributions of anthropogenic sectors (i.e., power plants, industry, transportation, and residential)."

Also note that as stated on page 3 line 14-17: "Top-down inversion typically provides the total emission data, although emissions from individual sources can be further derived by integrating a priori data (often from bottom-up inventories) about source-specific information such as diurnal and seasonal variabilities (e.g., Lin et al., 2010; Lin, 2012) and spatial variabilities (Timmermans et al., 2016)."

8.  Although the gridding is on a 0.05° resolution the actual spatial resolution of the resulting emissions seems much lower. An indication of the intrinsic resolution can be obtained from the largest gradients in the emissions. I cannot detect clear structures with a 0.05° resolution. One would at least expect some power plants to show up as clear spots in this region. Maybe the method is still limited by the OMI resolution?

The highest emission is at one grid cell in north Shanghai, and its difference from eight surrounding grid cells are 0.39 kg $km^{-2}$ $h^{-1}$ (2.6%). The mean gradient of the emissions is 0.079 kg $km^{-3}$ $h^{-1}$.

We admit the intrinsic resolution of our derived emissions is limited by the pixel sizes of OMI. We have added in the revised conclusion section that

On page 25 line 12-14: "The spatial resolution of the estimated emissions is limited by that of satellite VCD data, although a special oversampling technique has been used to help achieve the highest spatial resolution possible for emissions."

Also, in Sect. 4.2 on page 22 line 5-7: "This is because our top-down estimate is limited by the intrinsic resolution of $NO_2$ VCDs, i.e., our oversampling approach does not fully

compensate for the large sizes of OMI pixels. Therefore, the large spatial gradient of $NO_x$ emissions is smoothed to some extent in our dataset."

We have also discussed emissions related to power plants. As detailed in the revised Sect. 4.2 on page 20 line 10-18:

"The filled circles in Fig. 6g show the locations of coal-fired power plants in 2016 from Carbon Brief (www.carbonbrief.org; last access: 2019/6/27). The radius of a circle denotes the power generation capacity. Figure 6h further shows the GPED v1.0 bottom-up $NO_x$ emissions for power plants on a 0.1°×0.1° grid in 2016. Coal-fired power plants in the YRD are normally near the urban centers, traffic lines or other sources. Our top-down $NO_x$ emission map shows large emission values near the power plants (Fig. 6b), although it cannot isolate the sole contribution of power plants. At the GPED power plant locations, the correlation between our and GPED emissions reaches 0.26, due to the influence by non-power plant sources; note that the correlation between GPED emissions and POMINO NO2 VCDs are only about 0.21."

9. The structure of the paper is somewhat confusing and therefore I suggest moving appendix C and E to the main text. Section 2.1 is too short to understand the method of determining emissions.

We have substantially improved the structure of the manuscript to accommodate both reviewers' suggestions. A flowchart has been added to Sect. 2.1 in order to illustrate the procedures of our inversion method. Section 2.3 has been divided into 5 subsections for clarification. Section 2.3.1-2.3.3 describe the model setting and assumptions. Sect. 2.3.4 shows how the SCM matrix is applied to PHLET simulated VCDs, with the detailed procedures shown in Appendix B. Section 2.3.5 summarizes the uncertainty estimates. The part (former Appendix D) about solving the observation error covariance matrix and the adjoint model has been moved to Sect. 2.4, supplemented with an extended discussion on assuming the covariance to be diagonal. The OSSE-like test (former Appendix E) based on GEOS-Chem simulated $NO_2$ data has been moved to a new Sect. 5.

10. In section 2.3: The contributions of lightning, biomass burning and aircraft emissions are neglected. The authors explained that the contributions of these emissions are small. In the inversion method, both soil and anthropogenic emissions are derived. In section 4, it is calculated that the soil emissions contribute 0.9% of the inverted emissions. This looks like a very small amount. However, biomass burning is considered as a significant source in the YRD, especially in summer. On a scale of 0.05°×0.05° lots of biomass burning activity will exist. Give some more detailed information and explain why lightning, biomass burning and aircraft emissions are neglected.

As now clarified in Sect. 2.3.2 on page 10 line 13-21:

"Lightning emissions, biomass burning emissions, aircraft emissions, transport from neighboring regions, and convection can lead to $NO_2$ at higher altitudes over the YRD area. However, the amount of $NO_2$ aloft is much smaller than near-ground $NO_2$ due to large ground sources (Lin, 2012). Thus, we regard $NO_2$ aloft as the regional background, and do not include it in Eq. 1. Also, for near-ground $NO_2$ over the YRD area, the contribution of downward vertical transport is negligible compared to the contribution of ground sources. Aircraft emissions contribute little to the total ground source, because 78% of aircraft emissions occur at the high altitudes (9–12 km) (Ma and Xiuji, 2000). Therefore, PHLET only accounts for near-ground $NO_2$ from ground soil, biomass burning and anthropogenic sources (energy, industry, transportation, and residential)."

11. The model error is set to be the sum of the quadrature of errors contributed by several aspects. However, there is no explanation on how the authors set some errors, for example the treatment of background $NO_2$ concentrations. The authors use wind fields from ECMWF on a coarse resolution and regridded to a high resolution. The error of regridded wind field on high resolution can be quite large. The authors consider error of wind speed, but how about the wind direction? The error set for the wind looks optimistic.

Error estimates for individual parameters and processes are based on the literature, our sensitivity tests, and/or expert judgement. For most parameters, the reasoning of choosing specific values is given when the error terms are introduced.

For background $NO_2$, our choice ($0.54 \times 10^{15}$ molecules cm$^{-2}$) is based on the consideration that background $NO_2$ would be very small (e.g., smaller than $1 \times 10^{15}$ molecules cm$^{-2}$; Cui et al., 2016). Doubling the background only has marginal effects on our emission estimate especially at modest- and high-$NO_2$ locations. Spatially averaged, the error due to the choice of our background value is estimated as 5%.

For errors introduced by winds, we have clarified in Sect. 2.3.3 on page 11 line 17-19 that

"We assess the model errors introduced by the uncertainties in the wind field and effective diffusion coefficients by Monte Carlo simulations in which the wind speeds are changed according to their uncertainties. The resulting relative uncertainty in the modeled $NO_2$ VCDs is about 20%. "

12. References: All the references should be carefully checked if they are in the correct format, especially the names of authors. Many articles are missing or articles should be removed throughout the whole text. It is advisable to let a native speaker make the necessary corrections.

The references have been checked and some necessary citations have been added.

1. Page 1, Line 14: lacking => missing

Changed.

2. Page 1, Line 17: The inversion => The model used in the inversion

We have change it into 'the top-down inversion method' which refers to the whole process.

3. Page 1, Line 18: We construct a model called PHLET (..)

Changed.

4. Page 1, Line 19 Metrix => Matrix

Changed.

5. Page 2, Line 5 tied => related

Changed.

6. Page 2, Line 7 features => structures

Changed.

7. Page 2, Line 8-9: This last sentence is kind of obvious. It should be moved from the abstract to the conclusions/outlook, but I suggest to just remove it.

Removed.

8. Page 2, Line 21: split the sentence into 2 separate sentences to make it more understandable.

Changed. Now on page 2 line 18.

9. Page 2, line 24: on => of

Changed.

10. Page 2, line 24: how is a "very high resolution" defined?

We have made it clear to be 0.05°×0.05° on page 2 line 23 and page 4 line 9.

11. Page 3, Line 1: Bottom-up emissions do not only use spatial proxies but are also based on gathered statistical information of industrial output, car emissions, etc.

The statement has been modified.

We have made it clear to be 0.05°×0.05° on page 2 line 23 and page 4 line 9.

13. Page 3, Line 20-21: What do the authors mean by "low-cost" and "high-resolution"

Low cost refers to the low requirement on computation resources described on page 6 line 13-16:

"With one computational core (Intel ® Xeon ® Gold 6130 CPU @ 2.10GHz),

derivation of NOx emissions over the YRD here takes less than one hour after necessary input data are prepared. Applying the framework to multiple areas would take a similar amount of time by using one computational core for each area."

As for high resolution, we have made it clear to be 0.05°×0.05° on page 2 line 23 and page 4 line 9.

14. Page 3, Line 22: Here it is mentioned that these inventories are important for trends and variability. I agree, but the method presented in this paper do not give the possibility to study trends and variability, which should be mentioned somewhere in the conclusion.

As now clarified in Sect. 6 on page 24 line 22-24:

"Although this study derives the averaged emissions over summer 2012–2015, calculations of emissions at higher temporal resolutions (e.g., every 2 years) is possible to better capture the interannual variability and trends."

15. Page 3, Line 22-23: Why is it important to understand air pollution with the advent of TROPOMI? I would say it is the other way around: TROPOMI is important for understanding air pollution.

The sentence has been removed. Discussion about TROPOMI is on page 25 line 25-page 2 line 2 now.

Changed.

Changed. In addition, we have added two more cities which were missed in the original manuscript.

Changed. On page 6 line 22.

Changed.

Changed.

21. Page 4, line 17: Why is this the finest spatial information possible?

As described in Sect. 2.2 on page 7 line 25 – page 8 line 4, The oversampling approach takes advantage of the fact that the exact location of footprint of the OMI instrument slightly changes from one day to another, so does the exact location of footprint of a satellite pixel at a given VZA. Thus, sampling from multiple days increases the horizontal resolution of data. Besides, the SCM matrix is constructed base on the pixels, and thus the finest spatial information is preserved.

22. Page 4, line 19: What Is SCM? This is explained much later in the text.

We have re-structured the manuscript and have clarified the use of SCM in Sect. 2.3.4, supplemented by Appendix B.

23.  Page 4, line 21-22: Without further explanation this does not explain the method.

A flowchart has been added in order to illustrate the procedures of our inversion method on page 5 line 8 -page 6 line 4.

It refers to Eq. C5 in Appendix C. The statement has been modified.

25.  Page 5, line 1: Which nonlinear relationship do the authors mean here? There are 3 quantities mentioned: (1) emissions, (2) lifetimes and (3) VCDs.

Between lifetimes and $NO_2$ VCDs. Changed now on page 6 line 8.

26. Page 5, line 4: A long time period is mentioned. What do the authors mean, a long time period to average or multiple 5 years periods? And why are these long time periods not presented in this paper?

We meant summer 2012-2016.

We have shortened the study time period from summer 2012-2016 to summer 2012-2015. According to the National Bureau of Statistics of China (http://data.stats.gov.cn/), $NO_x$ emissions have dropped substantially from 2015 to 2016. Thus, including summer 2016 may not be the best practice to derive emissions.

27. Page 5, line 5: It is mentioned that the calculation takes about 36 hours after necessary input data? What are the necessary input data? How long does it take to prepare the input data?

The necessary input data are the OMI product (i.e. POMINO) and wind field. The time it takes to prepare those data depends on the Internet conditions, as in other studies. Running our codes to process these input data takes 30-40 minutes for the YRD domain here.

28. Page 5, line 5: If the inversion takes 36 hours for a 5x5 degree domain, a global calculation will take about 10000 hours, which is about 10 year.

See 'general comments 2'. And as stated on page 6 line 14-16, applying the framework to multiple areas would take a similar amount of time by using one computational core for each area.

29. Page 5, line 8: a reference for OMI is missing.

Added.

30. Page 6, line 2-7: Removing the 30 outer pixel and the row anomaly will strongly reduce the number of pixels used in this research. How may pixels be still used?

22007 pixels. Added.

31. Page 6, line 8: space => grid

Changed. Corresponding texts are now in Sect. 2.3.4

32. Page 6, line 11: The footprint does not change, the location of the footprint changes from one day to another.

Changed.

33. Page 6, line 14: the year of the reference to Fioletov is missing.

The sentence has been changed.

Changed, now on page 8 line 5.

35. Page 7, Line 7: The assumptions of the PHLET model are not mentioned in Beirle et al. This reference should be removed.

A similar assumption on the vertical shape of $NO_2$ is taken in Beirle et al. (2011), as their model does not include information about the (horizontal and temporal) changes in the vertical shape of $NO_2$. In their online supporting information, "At the OMI observation time under cloud free conditions, the megacity emissions undergo rapid vertically mixing (within some km distance from the source)".

36. Page 7, Line 10: The transport from neighbouring regions is missing in this list while this is an important contribution.

We consider the transport from outside the study domain as part of the regional background. We write in the revised Sect. 2.3.2 (page 10 line 13-16) that

"Lightning emissions, biomass burning emissions, aircraft emissions, transport from neighboring regions, and convection can lead to $NO_2$ at higher altitudes over the YRD area. However, the amount of $NO_2$ aloft is much smaller than near-ground $NO_2$ due to large ground sources (Lin, 2012). Thus, we regard $NO_2$ aloft as the regional background, and do not include it in Eq. 1."

37. Page 7, Line15-16: Can the authors give a reference for this statement.

The reference about aircraft emission is given. The statement has been adjusted since the biomass burning $NO_x$ should be taken into consideration.

38. Page 7, Line 17-20: This is quite some assumption about the background value. What is the basis of this assumption? Why is the uncertainty set to 5%?

For background $NO_2$, our choice ($0.54 \times 10^{15}$ molecules $cm^{-2}$) is based on the consideration that background $NO_2$ would be very small (e.g., smaller than $1 \times 10^{15}$ molecules $cm^{-2}$; Cui et al., 2016). Doubling the background only has marginal effects on our emission estimate especially at modest- and high-$NO_2$ locations. Spatially averaged, the error due to the choice of our background value is estimated as 5%.

Described in Sect. 2.3.3. The data are from ERA5.

40. Page 9, Line 20: space => grid

Changed.

41. Page 11, Line 7-8: I would suggest mentioning the average number of iterations (about 60?) needed to reach convergence and remove Fig C1. Is the value of 390 chosen based on this Figure and the fact that it is stable or are there other motivations?

There are 50 times of iterations before the convergence is reached according to the rate of decline of J. J is reduced from an initial value of 6585.2 to a stabilized value of 73.6.

We think the figure is important for the demonstration of how fast J is reduced. Thus, we have elected to keep the figure (Fig. 2).

42. Page 12, Line 6: It becomes clearer if the short appendix C is just put into the text here.

Adjusted.

43. Page 12, Line 11: "inverted emission" is not the correct term. The concentrations are inverted to get the emissions. This "emission inversion" and "inverted emissions" is appearing in many places in the text.

Changed throughout the text.

44. Page 12, Line 11-12: What is the value of error on the lifetime? I suggest mentioning also the values of the calculated errors in the text.

As described on page 15 line 11-14,

"The error in the lifetime is derived from the errors in $NO_x$ loss (estimated in Appendix C) and $NO_2$ VCDs, according to the common manner of error synthesis."

45. Page 14, line 1: inverted => derived

Changed.

46. Page 14, line 14: Since there is a lot of agriculture in the YRD region a soil contribution of 0.9% seems very small and needs some explanation. A discussion on biomass burning emissions (which occur in the agricultural regions) can be helpful here as well.

The soil emissions in GEOS-Chem already account for the effects of both fertilizer and natural soil. We cannot conclude whether the relative contribution of soil emissions (to the total) has been underestimated, because of the dominant emissions from power plants, transportation, industry and residential activities.

We have clarified that biomass burning is part of the sources of our derived emissions. In Sect. 4.2 where we compare our derived "anthropogenic" emissions with other anthropogenic inventories, the GFED v4 biomass burning inventory is adopted to be subtracted from our derived emissions.

47. Page 15, Line 1: Figure 2e is mentioned without discussing 2a-d.

Figure a-d have been discussed before.

Now we have re-arranged the figures. Additional figures have been added, considering comments from both referees.

48. Page 15, Line 13: Please mention the basis of the coloring of Fig 2f.

As stated on page 18 line 20-22, The data points are colored to indicate the different ranges of VCDs at individual grid cells.

49. Page 15, Line 19: Why are the emissions not directly compared to bottom-up inventories instead of these proxies that are used in the bottom-up inventories. For example, a comparison with the MarcoPolo-Panda or the Zhao et al. inventory can give more insights.

In Sect. 4.3 we compare our emissions with other inventories besides MEIC.

50. Page 17, Line 5-6: To separate the anthropogenic emissions, GEOS-Chem is used to calculate soil emissions. What are the uncertainties of the soil emissions calculated by GEOS-Chem?

We have discussed the errors in soil emissions and biomass burning emissions in Sect. 4.2 on page 21 line 13-19:

"Soil emissions are calculated by the nested GEOS-Chem (Fig. 7c), with the uncertainties assumed to be within 50% (Wang et al., 1998; J. Yienger and Ii Levy, 1995). Biomass burning emissions (Fig. 7b) are taken from the Global Fire Emissions Database (GFED4; www.globalfiredata.org/data.html; last access: 2019/7/10) (Giglio et al., 2013), with the uncertainties estimated to be within 10% over the YRD (Giglio et al., 2009; Giglio et al., 2013). Summed over the study domain, the soil sources contribute about 0.5% of our emissions while biomass burning contribute about 5.1%."

51. Page 17, Line 6: Figure 3ows?

Typo. Corrected.

52. Page 17, Line 8: Comparing Figure 3e and 3f is only useful if they are at the same resolution. Thus, Figure 3e should be regrided to the coarser resolution of Figure 3f.

All data are presented on 0.05°×0.05° grid.

53. Page 18, Lines 6-15: There is some repetition of the text of the previous sections

Rephrased.

54. Page 18, Line 19-20: I would remove the last sentence about the programming language, which is not very relevant in a scientific paper

We have elected to keep the sentence, because we consider that programing with Python, a popular and easily used language, is an important feature that the potential users of our codes may appreciate.

55. Page 22: Line 17: Since one observation of the satellite is used in several grid cells, I doubt if the assumption that covariance matrices are diagonal matrices is correct. A discussion is needed here.

Since we use several pixels to get the mean VCD for each grid cell, and grid cells nearby each other may shares the same pixels partly although weight differently. Therefore, we admit that making the covariance matrix to be diagonal may be an imperfect assumption, although a similar assumption has been used in many previous studies (Keiya and Itsushi, 2006; Cao et al., 2018).

To partly account for the uncertainty lead by such approximation, we assume relatively high errors in the VCDs, as shown in Sect. 2.2:

On page 8 line 5-13: "For the purpose of emission estimate, we assume that the error of VCD at a satellite pixel ($\sigma_p$) contains an absolute error of half of the mean VCD over the domain (i.e.,$1.9\times10^{15}$ molecules cm$^{-2}$) and a relative error of 30% (Lin et al., 2010;Boersma et al., 2011;Lin et al., 2015a;Beirle et al., 2011). We further add in quadrature an additional error ($\sigma_g$) when a satellite pixel is projected to the grid cells at a finer resolution; this error is important in the urban-rural fringe zone. For a given grid cell, $\sigma_g$ is set to be 50% of the standard deviation of VCDs at its eight surrounding grid cells. Sampling over multiple days reduces the random error by a factor of $s = \left(\sqrt{(1-c)/n + c}\right)$ , where c represents the fraction of systematic error (assumed to be 50%) and $n$ the number of days with valid data (Eskes et al., 2003; Miyazaki et al.,

2012). Thus, the total error for the temporally averaged VCD at a given grid cell is

$$\sigma_s = \sqrt{\left(\sigma_p^2 + \sigma_g^2\right) \cdot s} \ . "$$

We have also summarized the limitation in the conclusion section:

On page 25 line 16-18: "The adjoint model assumes the observational error covariance matrix to be diagonal, without fully considering the effect of correlations between individual grid cells."

56. References: Most references contain many spelling errors and omissions.

Checked.

57. Figure 2d: The lifetime is very short over the ocean, contradicting to what is usually seen in the literature.

We agree that the lifetime over the ocean might be underestimated and the emission there is therefore overestimated. Some discussions are added on page 17 line 6-11 and on page 25 line 12-13.

58. Figure 2e: Although the gridding is on 0.05 degree resolution the actual spatial resolution of this image seems much lower. An indication if the intrinsic resolution can be obtained from the largest gradients in the emissions. I cannot detect clear structures with a 0.05 degree resolution. One would at least expect some power plants to show up as clear spots in this region.

Maybe the method is still limited to the OMI resolution. I would like to see some discussion about this.

Please see our response to general comment 8.

59. Figure 2f: The plot is more logical when the x-axis and y-axis are reversed. I also suggest drawing a line for the 100% relative error in this plot as a helpline to guide the eye.

Changed.

60. Figure 2 caption: What are the magnitudes of POMINO that are mentioned. In Figure 2f it is too small to see.

Changed.

61. Figure 3e: This has to be regrided to the resolution of 2f for comparison.

Adjusted.

62. Figure 3: I miss information on power plants, which are a major source of $NO_x$.

See our response to general comment 8.

63. Figure 4: The crosses in the plot, indicating the amount of grid cells, are too small.

The crosses have been deleted for simplicity.

64. Figure D1: I do not understand why we need two colors for the dots.

The blue points stand for the grid cells where the VCDs is lower than $5 \times 10^{15}$ molecules cm$^{-2}$, and the evaluated local net sources at these grid cells are used to derive the relation between the $NO_x$ loss and $NO_2$ VCDs by fitting the fixed formula. An explanation is added on page 30 line 7.

65. Figure D2: The lifetime depends a lot on chemistry, temperature and precipitation. Therefore, the plot seems very simplified.

We agree, but this is the best we can do without involving a computationally much costlier 3-D chemical transport model. We note that previous studies with simplified models have often assumed a single value for lifetime for each city or other emission sources (Beirle et al., 2011; Liu et al., 2016).

We have added a discussion of the limitations of our method, including about the lifetime, in the revised conclusion section on page 25 line 18-20:

[revised manuscript text omitted]

---

## Referee Report (RR1)

**Review "High-resolution (0.05°x0.05°) NOx emissions in the Yangtze River delta inferred from OMI"**

Since the previous review, many changes and additions have been made to the manuscript. This has improved the manuscript a lot. A few additions I consider not valuable enough for the main text and maybe can be moved to the supplementary material. Section 4.2 in which the emissions are compared with spatial proxies is rather superficial and not very quantitative. Not much can be concluded from it and I suggest to move this to the supplementary material. Section 5 is not an OSSE comparison, but more a test for internal consistency. Thus the naming should be changed and I suggest to move this also to the supplement.

The manuscript still contain many grammar errors , missing articles (ie. "the" and "a") and the references have many mistakes. Below I mention several of them.

Page 3, Line 8: remove first reference to Zhao et al

Page 3, Line 10: change to demanding, and therefore difficult to be applied …

Page 5, Line 11 What are pixel parameters ?

Page 8, line 5 "estimate" =>" estimating"

Page 8, Line 20 "based on" => "by"

Page 9, Line 17: "become" => "because"

Page 10, Line 12: "as" => " to"

Page 12, Line 6: "abovementioned" => " above mentioned"

Page 17, Line 13: "based on the" => " as can be concluded from "

Page 21, Line 8: The reference to Timmermans et al. seems not correct and redundant.

Page 24, Line 20: The use of a Google photo is not the equivalent to a validation with land use data. Either use a real land-use database or just remove the comparison to GoogleEarth from the manuscript.

Page 25, Line 22: I remain of the opinion that the programming language used in this study is not relevant to be mentioned in an ACP manuscript and should certainly not be mentioned in the concluding remarks.

I noticed mistakes or omissions in the references Boersma-2011, Ding-2017a,Elvidge-2013, Hersbach-2016, Yineger-1995, Liu-2017, Mahieu-2013, Mijling-2012, Sun-2018, Timmermans-2016

Figure 2: This is a rather textbook figure of a cost function and I suggest to remove it.

Figure 7: Images (c) and (e) have interpolated colors at the edges of the grid boxes.

---

## Author Response (AR2)

1.      There are a few grammatical errors (e.g. Page 1, Line 16, to estimating), please find a native speaker to proofread the paper.

Changed.

5  2.      The quality of the figures by increasing both the resolution and the sizes of the figure legends need be enhanced.

The legend of revised Fig. 8 has been enhanced. I am afraid that it may be impossible to improve the resolution of the revised Fig. 6, since Fig. 6f and 6g are captured from the cited website, and the raw data are not available. We have improved the quality of the other figures.

**10  Anonymous Referee #2**

General Comments

Since the previous review, many changes and additions have been made to the manuscript. This has improved the manuscript a lot. A few additions I consider not valuable enough for the main text and maybe can be moved to the supplementary material. Section 4.2 in which the emissions are compared
15  with spatial proxies is rather superficial and not very quantitative. Not much can be concluded from it and I suggest to move this to the supplementary material. Section 5 is not an OSSE comparison, but more a test for internal consistency. Thus the naming should be changed and I suggest to move this also to the supplement. The manuscript still contain many grammar errors, missing articles (i.e. "the" and "a") and the references have many mistakes. Below I mention several of them.

20  We choose to keep Sect. 4.2 and Sect. 5. We compare our emissions with the proxies such as population and nighttime light brightness to substantiate the spatial pattern of our emissions which may not be captured by other inventories as shown in Sect. 4.3. Besides, readers may be interested in the relation between those proxies and $NO_x$ emissions. And we think it better to keep Sect. 5 in the main text, taking

your previous suggestion into consideration. This section offers another independent method to substantiate the robustness of our inversion. Therefore, we keep those two parts.

We have checked for the grammar errors and missing articles.

Specific comments

5    3.  Page 3, Line 8: remove first reference to Zhao et al

Removed.

4.  Page 3, Line 10: change to demanding, and therefore difficult to be applied …

We have changed this sentence into "on-site surveys are extremely time consuming and resource demanding, and therefore difficult to be applied to a large domain in a timely manner."

10   5.  Page 5, Line 11 What are pixel parameters ?

The pixel parameters refer to the corner co-ordinates. We have made a brief explanation in the revised paper.

6.  Page 8, line 5 "estimate" =>" estimating"

Changed.

15   7.  Page 8, Line 20 "based on" => "by"

Changed.

8.  Page 9, Line 17: "become" => "because"

Changed.

9. Page 10, Line 12: "as" => " to"

Changed.

10. Page 12, Line 6: "abovementioned" => " above mentioned"

Changed.

5   11. Page 17, Line 13: "based on the" => " as can be concluded from "

Changed.

12. Page 21, Line 8: The reference to Timmermans et al. seems not correct and redundant.

We have removed the reference here.

13. Page 24, Line 20: The use of a Google photo is not the equivalent to a validation with land use data.
10      Either use a real land-use database or just remove the comparison to GoogleEarth from the manuscript.

We choose to keep this comparison, since this satellite photo from GoogleEarth taken by Landsat, which
has been sufficiently used in research on land use, presents a rather intuitive indication of land use and
overall human activities, which account for a major part of anthropogenic $NO_x$ emissions.

14. Page 25, Line 22: I remain of the opinion that the programming language used in this study is not
15      relevant to be mentioned in an ACP manuscript and should certainly not be mentioned in the
        concluding remarks.

We still choose to keep this description, because we consider that programing with Python, a popular and
easily used language, is an important feature that the potential users of our codes may appreciate.

20   15. I noticed mistakes or omissions in the references Boersma-2011, Ding-2017a, Elvidge-2013,

Hersbach-2016, Yineger-1995, Liu-2017, Mahieu-2013, Mijling-2012, Sun-2018, Timmermans-2016

Changed.

16. Figure 2: This is a rather textbook figure of a cost function and I suggest to remove it.

We still choose to keep this figure. The cost function certainly decreases in a similar manner when an adjoint method is adopted. However, the rate of descending and the amount of iteration times differ from case to case. Therefore, we keep this figure.

17. Figure 7: Images (c) and (e) have interpolated colors at the edges of the grid boxes.

As stated on page 19 line 9,

"When compared with our emissions, these emission data are regridded to 0.05°×0.05°."

And the lattices of the grids do not match well, and thus, when regridded to our grid, the data have to be interpolated.

**Marked-up manuscript**

[revised manuscript text omitted]